

# Prediction of the area affected by earthquake-induced landsliding based on seismological parameters

Odin Marc[1], Patrick Meunier[2], and Niels Hovius[1, 3]

[1]Helmholtz Centre Potsdam, German Research Center for Geosciences (GFZ), Telegrafenberg, 14473 Potsdam, Germany.
[2]Ecole Normale Superieure de Paris, Laboratoire de Geologie, 75231 Paris CEDEX 5, France
[3]Institute of Earth and Environmental Science, University of Potsdam, Potsdam, Germany

*Correspondence to:* Marc O. (odin.marc@unistra.fr). Now at Institut de Physique du Globe de Strasbourg, CNRS UMR 7516, University of Strasbourg, 5 rue Descartes, F-67084 Strasbourg, France

**Abstract.** We present an analytical, seismologically consistent expression for the surface area of the region within which landslides induced by a given earthquake are distributed. The expression is based on scaling laws relating seismic moment, source depth and focal mechanism with ground shaking and fault rupture length and assumes a globally constant critical acceleration for onset of systematic mass wasting. The seismological assumptions are identical to those recently used to propose a seismologically consistent expression for total landslide volume and area. To test the accuracy of the model we gathered geophysical information and estimates of the landslide distribution area for 83 earthquakes. To reduce uncertainties and inconsistencies in the estimation of the landslide distribution area, we propose an objective definition based on the shortest distance from the seimsic wave emission line containing 95% of the total landslide area. Without any empirical calibration the model explains 56% of the variance in our dataset, and predicts 35 to 49 out of 83 cases within a factor two, depending on how we account for uncertainties on the seismic source depth. For most cases with comprehensive landslide inventories we show that our prediction compares well with the smallest region around the fault containing 95% of the total landslide area. Aspects ignored by the model that could explain the residuals include, local variations of the critical acceleration and processes modulating the surface ground shaking, such as the distribution of seismic energy release on the fault plane, the dynamic stress drop or the rupture directivity. Nevertheless, its simplicity and first order accuracy suggest that the model can yield plausible and useful estimates of the landslide distribution area in near-real time, with earthquake parameters issued by standard detection routines.

## 1 Introduction

Triggered landslides are a significant secondary hazard of earthquakes, and may be the dominant cause of damage to infrastructure and lifelines, especially roads (Bird and Bommer, 2004). The severity of this hazard and the associated risks is clear after most large earthquakes in steep landscapes, and was underlined by the devastation and fatalities caused by landsliding induced by recent earthquakes in Sichuan (China) 2008 and central Nepal, 2015 (Yin et al., 2009; Kargel et al., 2015). The earthquake-induced landslide hazard is defined in the first instance by the number, size and location of landslides. These variables are correlated with a combination of local factors, such as the peak ground acceleration (Meunier et al., 2007, 2008), hillslope geometry (Parise and Jibson, 2000; Gorum et al., 2013), and the strength (Parise and Jibson, 2000; Gallen et al.,





2015) and degree of saturation of near-surface materials, which are difficult to quantify due to their inhomogeneity across

epicentral areas (Dreyfus et al., 2013). A simpler approach is to predict first-order variables such as the total volume and area of landsliding caused by an earthquake based on simple seismological considerations (e.g., Marc et al., 2016b). If the input parameters for such a model can be quantified in minutes to hours after an earthquake, then this could yield a quick insight into the scale of the affected area and the total amount of landsliding within. Here we focus on the landslide distribution area, $A_d$, that is the surface area of the region within which landslides induced by a given earthquake are likely to be concentrated. This

is an important risk parameter, as it defines the zone within which the landslide hazard is focused. It can be intersected with areas of vulnerability for an assessment of risk and defines the extent of the area experiencing seismically-induced hillslope denudation, with geomorpohological (e.g., Marc et al., 2016a), geochemical (e.g., Jin et al., 2015), tectonic (e.g., Steer et al., 2014) and biological consequences (e.g., Garwood et al., 1979).

Several global or regional compilations of earthquake-induced landslide data have reported $A_d$, and explored its scaling with

20 earthquake magnitude (Keefer, 1984; Rodriguez et al., 1999; Hancox et al., 1997). They found that $log_10(A_d)$ scales linearly with the moment magnitude $M_w$ of an earthquake, or:

$$A_d \sim Mo^{2/3}, \tag{1}$$

where $Mo$ is the earthquake seismic moment. Because the scatter around the central trend of this relation is substantial, a common approach is to base a prediction of $A_d$ on the envelope defining the maximum area affected by landsliding for a given

earthquake magnitude. Further, some case studies have illustrated how some seismic or geomorphic conditions can lead to landslide triggering over exceptionally long distances and therefore large areas (Keefer, 2002; Jibson and Harp, 2012). However, the definition of the relationship between $A_d$ and seismic moment and of the influence of other seismological and geomorphic parameters has a slender theoretical basis. Recently, expressions for the total area and volume of earthquake-induced landslides have been derived based on seismological scaling laws and simple topographic characterizations (Marc et al., 2016b).

Here we show that this same treatment may be used to predict the shape and size of the landslide distribution area. First, we present the basis for an expression of the landslide distribution area and the landslide maps and compilations of estimated landslide distribution areas against which model predictions can be compared. Then we assess the validity and accuracy of our theoretical approach and discuss its limitations, and finally we suggest directions for future improvements.

## 2 A seismologically-consistent expression for the landslide distribution area

Earthquakes trigger landsliding due to transient accelerations during ground shaking, which shifts the force balance in a slope and causes damage in the substrate, reducing its cohesion and resistance to failure (Newmark, 1965).

This link between landsliding and ground shaking, specifically peak ground acceleration, is confirmed by a growing number of detailed observations (Khazai and Sitar, 2004; Meunier et al., 2007; Yuan et al., 2013). They constrain, for example, the statistical occurrence of landsliding within areas where the ground acceleration normalized by the gravitational acceleration



exceeds a critical threshold, $a_c \sim 0.1 - 0.2$ (Meunier et al., 2007; Hovius and Meunier, 2012; Yuan et al., 2013). Below this threshold, landsliding is rare or minor, and therefore a reasonable estimation of the size and shape of the landslide distribution area could be found by intersecting the region where the peak ground acceleration exceeds $a_c$ and the region with sufficiently steep topography for landsliding to occur.

Marc et al. (2016b) have successfully modeled the total volume and area of the population of landslides due to a given earth-
quake, using seismological scaling laws to constrain the magnitude and extent of ground shaking. They assumed that $a_c$ was the acceleration at which damage and an effective reduction of strength occurred within the hillslope materials, increasing the likelihood of failure of oversteepened slopes, and used $a_c = 0.15 \pm 0.02$. Note, that according to this assumption $a_c$ should depend only on the material properties and be independent of the hillslope gradient. Their model considers attenuation of seismic waves due to geometric spreading, with S-wave amplitude decreasing with distance between the earthquake source and the
affected topography. The pattern and intensity of ground shaking are modeled as the superposition of point sources associated with asperities along the fault rupture. Therefore, the source of wave emission is considered to be distributed along the fault rupture length, while the wave acceleration at the source, that sets the distance over which waves can travel before becoming insufficient to trigger landslides, scales non-linearly with the magnitude of the earthquake. With the same assumptions we can predict the area affected by ground shaking exceeding $a_c$, $A_s$:

$$A_s = 2LR_{HMAX} + \pi R_{HMAX}^2, \tag{2}$$

where $R_{HMAX}$ is the horizontal distance from the surface projection of the wave source at which the ground shaking reaches $a_c$. Assuming that wave attenuation is dominated by geometric spreading and that non-linear attenuation can be neglected, $R_{HMAX} = \sqrt{(b/a_c)^2 - R_0^2}$, with $b$ the acceleration inferred at 1km from the seismic source and $R_0$ the mean depth of seismic wave emission. The assumptions are justified near the fault where most landslides occur (Marc et al., 2016b), but additional
non-linear attenuation would result in a reduction of the predicted $R_{HMAX}$ and $A_s$. The mean depth of emission is assumed to be the mean asperity depth because asperities emit most of the high frequency waves (Ruiz et al., 2011; Avouac et al., 2015) and seem to explain best the observed patterns and amounts of landsliding (Meunier et al., 2013; Marc et al., 2016b). Note that waves with frequencies of 0.5 to 10Hz are often the most important for landslide triggering because they have wavelengths ranging from the landslide size to the hillslope size (Marc et al., 2016b).

Following Marc et al. (2016b), we use the scaling of fault rupture length, $L$, with seismic moment proposed by Leonard (2010):

$$L = \frac{Mo^{2/5}}{\mu C_1^{3/2} C_2}, \tag{3}$$

with $\mu$ the shear modulus, assumed to be 3.3 GPa, and $C_1 = 16.5 m^{1/3}$ and $C_2 = 3.7.10^{-5}$ constants derived empirically from many earthquakes (Leonard, 2010). And we use the scaling of the near-source acceleration, $b$, with magnitude proposed by Boore and Atkinson (2008):

$$b = b_{sat} \exp\left(e_5(M_w - M_h) + e_6(M_w - M_h)^2\right) \tag{4}$$

$$\forall M_w > M_h, \quad b = b_{sat} \exp\left(e_7(M_w - M_h)\right)$$





where $M_h = 6.75$ is a hinge magnitude above which the acceleration carried by seismic waves saturates near $b_{sat}$, and $e_5 = 0.6728$, $e_6 = -0.1826$ and $e_7 = 0.054$ are empirical constants for 1Hz waves (Boore and Atkinson, 2008). Following Marc et al. (2016b), we use $b_{sat} = 4000m$, yielding surface acceleration of 0.4-0.8 above asperities located at 5-10 km depth,

consistent with observations during earthquakes with $M_w > M_h$. Fault type may influence both rupture length and ground shaking. For the shaking term, we follow the model choices of Marc et al. (2016b), attributing 30% less shaking for normal fault earthquakes than for equivalent strike-slip or reverse slip events (Boore and Atkinson, 2008). Large earthquake ruptures on strike-slip faults often break the seismogenic layer over its entire depth, fixed to 17km in our model (Leonard, 2010). There-fore, above a critical magnitude the increase of fault rupture area with seismic moment is only due to fault rupture length and

the scaling exponent between moment and rupture length becomes 2/3. Although, a similar scaling break may exist in principle for dip-slip faults it is less clearly observed (Leonard, 2010) and therefore not prescribed in our model.

With these considerations, both the fault rupture length and near-source wave amplitude can be estimated from the seismic moment and fault type. Thus Eq.2 can be rewritten more explicitly and corrected for the presence of topography with a topographic index, $C_{topo}$, to obtain the predicted area affected:

$$25 \quad A_{dp} = C_{topo}A_s = 2\,C_{topo}\left(L(Mo,F)\sqrt{\left(\frac{b(Mo,F)}{a_c}\right)^2 - R_0^2} + \pi\left(\left(\frac{b(Mo,F)}{a_c}\right)^2 - R_0^2\right)\right), \quad (5)$$

Hence, the landslide distribution area can be predicted from the earthquake moment and mechanism, and the mean asperity depth for any earthquake. If $a_c$ is relatively constant as suggested by Marc et al. (2016b) then the model is fully constrained without any free parameter. This assumption that $a_c$ is constant across all settings is discussed in Section 5.3. Note that $C_{topo}$ cannot be computed as the fraction of $A_s$ within which slopes are less than 10°because local flats will impede landsliding

and change the landslide density, but rarely the distribution area defined as an envelope containing all earthquake-induced landslides, independent of variations of landslide density. Only large flat areas, extending beyond $R_{HMAX}$ and with a width similar to the fault length will matter. Typically the presence of a basin or inundated areas along the entire fault will half the landslide distribution area ($C_{topo} = 0.5$).

In the model, the critical seismic moment, above which $A_s$ assumes a non-zero value is modulated by $R_0$ and ranges between $10^{16} - 10^{19} N.m$ (Figure 1). Above this critical moment, $A_s$ rises sharply, driven by the exponential increase of the source acceleration ($b$) with increasing moment (Eq 4). Upon reaching the hinge magnitude, $M_h = 6.75$, $b$ saturates and $A_s$ increases

5 primarily due to increase of the fault rupture length ($L$) with moment. Therefore, for these large events $A_s$ scales as a power-law of the moment, with an exponent of 2/5 for dip-slip events, and an exponent of 2/3 for strike-slip events (Figure 1). For large events, $b/a_c$ is about 27km and therefore $R_{HMAX}$ is almost independent of $R_0$ unless the latter reaches depths $> \sim 20km$. Thus uncertainties on $R_0$, which is the least well-constrained input variable of Eq. 5, will not substantially affect predictions of the landslide distribution area for most large, shallow earthquakes that rupture the upper crust (Figure 1). However, for some

10 intermediate size earthquakes or earthquakes deeper than 25km, the prediction may vary dramatically due to minor changes in the estimated source depth.



## 3 Data and methods

Our model predicts the landslide distribution area $A_d$, and its first order shape. However, for many earthquakes we have poor constraints on the shape and definition of landslide distribution area. Therefore, we test the model in two different ways. First, we use a large database of earthquakes containing geophysical information and a, sometimes crude, estimate of $A_d$, to assess if the scaling in the model matches the data and appears to yield a correct first-order prediction of $A_d$. Secondly, we focus on a limited number of cases for which we have a detailed landslide inventory that allow us to define a simple and objective parameter to characterize $A_d$ and to compare it quantitatively to the model prediction.

### 3.1 Landslide maps and compilations of landslide distribution area

To assess if our theoretical framework captures the observed scaling between $A_d$ and $Mo$, we test it against a large database of 83 crustal earthquakes, for which magnitude and location can be reasonably well constrained. These 83 cases have been harvested from published compilations (Keefer, 1984; Hancox et al., 1997; Rodriguez et al., 1999; Bommer and Rodriguez, 2002; Martino et al., 2014) and from recent landslide maps (Table 1). They include the 10 cases with comprehensive landslide inventories described separately below, 36 inventories for which we could access one or several maps with isolines of landslides density or point inventories to check the values reported in published compilations (Bonilla, 1960; Keefer et al., 1980; Harp et al., 1984; Harp and Keefer, 1990; Jibson et al., 1994; Tibaldi et al., 1995; Hancox et al., 1997; Keefer and Manson, 1998; Hancox et al., 2003, 2004; Jibson and Harp, 2006; Mahdavifar et al., 2006; Sato et al., 2007; Kamp et al., 2008; Mosquera-Machado et al., 2009; Alfaro et al., 2012; Collins et al., 2012; Jibson and Harp, 2012; Gorum et al., 2014; Martino et al., 2014; Xu et al., 2014a, b, 2015; Martha et al., 2016; Zhou et al., 2016), and a further 37 cases for which we could not access any raw data to evaluate the reported values (Table 1).

For ten earthquakes, detailed landslide inventories with comprehensive maps of the landslide as polygons are available, allowing an objective characterization of $A_d$ (as discussed below): the 1976 Guatemala, 1991 Limon, 1993 Finisterre, 1994 Northridge, 1999 Chi-Chi, 2004 Niigata, 2007 Aysen, 2008 Iwate, 2008 Wenchuan and 2010 Haiti earthquakes, ranging from Mw 6.2 to 7.9, and with hypocentre depths between 3 and 25 km (Table 1, (Harp et al., 1981; Harp and Jibson, 1996; Liao and Lee, 2000; Yagi et al., 2007; Meunier et al., 2008; Yagi et al., 2009; Gorum et al., 2013, 2014; Xu et al., 2014c; Marc et al., 2016b)). Most of these inventories were produced from extensive imagery and include all landslides that could be detected at the full image resolution. Two inventories deviate from this. The 1976 Guatamala inventory is based on high-resolution airphotos, but only covers a limited areas containing the most intense landsliding. Published values of $A_d$ for this case are larger than our estimate from the landslide inventory, considered as a lower bound. However, the inventory for Guatemala is considered to contains more than 90% of the landslides triggered by the 1976 earthquake (Jibson, pers. comm. 2013). Landslides triggered by the 1991 Limon earthquake were mapped across a wide swath Landsat-5 image and the limit of the disturbed areas could be constrained, but the low image resolution (30m) did not allow the delineation of all individual landslides in the most intensely affected area. Therefore, the distribution area $A_d$ is probably not significantly underestimated, but the cumulative surface area of landslides within it is, and any calculations based on that measure may be biased (cf., Marc et al., 2016b). We note that for



some earthquakes such as the Wenchuan and Haiti events, several inventories are available. For the Wenchuan earthquake the inventory by Xu et al. (2014c) seems the most comprehensive and robust, compared to earlier mapping. For Haiti we analyze and compare the inventories of Gorum et al. (2013) and Harp et al. (2016), which have different interpretations in some areas, likely due to differences in the pre and post earthquake imagery used. Most of the ten landslide inventories are not complete because they are limited by the resolution of the available imagery, and do not include very small landslides and rockfalls that can be detected in the field (e.g., Jibson and Harp, 2012). As a result these inventories can yield $A_d$ estimates smaller than those from previous compilations, but our estimates may be more representative of the area affected by dense landsliding, which is of primary interest in terms of both hazard and erosion. For most earthquakes we have information about moment magnitude, hypocentral depth and focal mechanism from the international seismological center catalogue (Storchak et al., 2013), but no estimate of the mean asperity depth most relevant to describe the mean wave emission depth (Table 1). However, we have shown that for large earthquakes, $A_{dp}$ is not very sensitive to depth, while for small earthquakes we expect the hypocentral depth and mean asperity depth to be close of each other. $C_{topo}$ was crudely estimated based on the topography and the fault position, and is about 1 for most cases (68 out of 83), about 0.5 for 10 events (coastal/basin geometry) and less than 0.5 for 5 cases (Table 1). For a small fraction of cases in our database we could find a stress drop estimate (Allmann and Shearer, 2009; Baltay et al., 2011) (Table 1), which must correlate with the wave emission at the source and thus the ground shaking intensity (Hanks and McGuire, 1981; Baltay and Hanks, 2014). All stress drop estimates were converted to an equivalent dynamic stress drop as defined by Brune (1970). Subduction earthquakes and distant offshore earthquakes were ignored because the area affected by strong shaking is mostly submerged and hillslopes are only present at large distance where the shaking intensity may not be well approximated by our simple model (cf., Marc et al., 2016b).

## 3.2 An objective definition of the landslide distribution area

It is important to secure consistency between estimates of $A_d$ from different sources and to constrain the degree of uncertainty associated with these estimates. Commonly, the landslide distribution area is estimated by locating all landslides caused by an earthquakes as accurately and comprehensively as possible, and drawing a single, smooth envelope containing all landslides, regardless of possible variations of landslide density within it (Keefer, 1984; Hancox et al., 1997; Rodriguez et al., 1999). However, many published inventories are limited by the spatial extent and quality and resolution of the available imagery, and may not include the small landslides in the far field, which, if taken into account, would give rise to a much greater $A_d$. For example, in the case of the 2008 Wenchuan earthquake, accounting for sparse landsliding in the far field would give $A_d \sim 200,000 km^2$ instead of $A_d \sim 44,000 km^2$ (Xu et al., 2014c). It is also likely that for large earthquakes with widespread landsliding, there is a tendency to focus on the most intensely affected areas, while for smaller earthquakes, the extent over which small rockfalls occur may be investigated in greater detail through field investigations (Jibson and Harp, 2012, 2016). For small to medium earthquakes triggering a relatively limited number of landslides erroneous inclusion of landslides triggered by other processes just before or after the main shock may also cause a significantly upward bias of $A_d$ estimates. In most cases we lack the information required to assess the accuracy and consistency of $A_d$ estimates between different events, but we note that using the common method described just above to determine $A_d$ from published maps or detailed inventories, we could reproduce within



$\sim 20\%$ $A_d$ estimates reported in global compilations and citing the same source study (Keefer, 1984; Rodriguez et al., 1999). For 27 earthquakes, we found different estimates of $A_d$, from different publications, methods and/or source imagery.These include different $A_d$ reported when considering only the area affected by intense landsliding, or including more distant, sparse landsliding (e.g., Hancox et al., 1997; Xu et al., 2014c). This is not an adequate quantification of the uncertainties in the dataset as a whole, but serves to illustrate how estimates of Aa may vary or be biased (Figure 1 Inset).

We propose a robust, alternative approach to define $A_d$, based on the fault rupture and the landslide inventory. In this approach, $A_d$ is defined as the surface area of the region within a distance $R_{95}$ from the seismic wave emission line projected at the surface. This region is set to contain 95% of all landslides triggered by an earthquake, as measured by their surface area. In this definition, $R_{95}$ is a 1D measure of the spread of landsliding away from the seismic source. The source is modeled as a wave emission line (or series of lines), defined by the location and length of the earthquake rupture, as determined by geophysical inversion of the slip distribution (or moment distribution for the Guatemala earthquake) on the seismogenic fault plane (Figure 2, 3, (Kikuchi and Kanamori, 1991; Wald et al., 1996; Zeng and Chen, 2001; Hikima and Koketsu, 2005; Hayes et al., 2010; Suzuki et al., 2010; Fielding et al., 2013). The rupture length of such inversions agrees (within 30% or less) with the a-priori predicted length for the Chi-Chi, Haiti and Northridge earthquakes, but is smaller (by 40-50%) for the Niigata and Iwate earthquakes, and larger for the Wenchuan and Guatemala earthquakes (Table 1). This results in uncertainties when attempting to measure or predict $R_{95}$ without knowledge of the rupture length and position, for example for old earthquake or just after an earthquake has occurred. For the Finisterre case we only have the position of the epicenter, the fault strike and the aftershock locations. They define a long fault rupture with the main shock, $M_w 6.9$, being closer of the northwestern fault tip and a large secondary shock, $M_w 6.5$ farther east (Stevens et al., 1998). Accordingly, we defined two separate emission lines, both with an epicenter located 1/3rd of the rupture length from the respective tips (Figure 3). For the Limon and Aysen earthquakes we placed the emission line centered on the maximum of moment emission and epicenter, respectively (Goes et al., 1993; Legrand et al., 2011), and oriented according to the focal mechanism. In the latter case, choosing the alternative focal mechanism (90° rotation) would not change significantly our results.

Our treatment differs from previous definitions of the maximum distance for landsliding (e.g., Keefer, 1984; Hancox et al., 1997; Rodriguez et al., 1999), because we consider a seismic line source that may be offset from the surface rupture of the seismogenic fault, and because it is not based on individual detected landslides but on the distribution of landsliding. Advantages of using the $R_{95}$ criteria as compared to previous approaches are its objectivity and reproducibility, and its robustness to the accidental addition or omission of minor landsliding in the far field. Thus, $R_{95}$ is strongly related to the seismic forcing and still representative of the area where hazard and erosion are likely to be most significant. A drawback is that this approach requires polygon inventories. Using 95% of the landslide number as a threshold for point based inventories would be an adequate solution but this definition would still be quite sensitive to effects affecting the number of identified landslides such as the imagery resolution and/or amalgamation of adjacent smaller landslides (Marc and Hovius, 2015). Another drawback is that $R_{95}$ assumes equal rates of decrease of landslide density with distance from the emission line in all directions, which may not always be the case.





## 4 Results

### 4.1 Comparison of observed and predicted landslide distribution areas

Most of our landslide distribution area data fall within the range of theoretical predictions from Eq 5, based only on fixed global parameters and variable earthquake settings (Figure 1). In Figure 4, the predicted distribution areas, $A_{dp}$, calculated accounting for the hypocentral depth and adjusting for the abundance of hillslopes where landslides may occur ($> 10°$)(Marc et al., 2016b), are plotted against values estimated from observations, $A_d$, for the earthquakes in our database. For 42% of 83 earthquake cases, Eq 5, yields $A_{dp}$ within a factor 2 of $A_d$ when considering the hypocentral data as the exact emission depth ($R^2 = 0.56$). This increase to 59% of the events when setting the emission depth within 25% of the hypocentral depth (Figure 4, inset). Landslide distribution areas vary in size between 10 and $10^5 km^2$, but half of the earthquakes have $A_d$ between $10^3$ and $10^4 km^2$. In this range, the predictions are mostly within a factor of 2 from the estimated area. When constrained, uncertainties on $A_d$ estimates are within a factor of 2 and 4 for about 2/5th and 4/5th of the well constrained cases, respectively, but occasionally the ratio between $A_d$ estimates from different sources may reach up to a factor of 10 (Figure 1 inset). For a number of small to moderate magnitude earthquakes with a small $A_d$, the model predicts no landsliding. This maybe due to uncertainties on the hypocentral depth estimation that we assume to be the depth of wave emission, $R_0$, and assigning a $< 25%$ uncertainty on $R_0$ allows for the prediction to match $A_d$ within a factor of 2, for 5 out of 7 cases in this category (Figure 4). Most earthquakes with $A_d > 10^4 km^2$ have $M_w > 7.5$ and are shallow enough ($R_0 < 20km$) to be relatively insensitive to depth. Nevertheless, they are often underpredicted by our model by a factor 1.5 to 3 (Figure 4). Notwithstanding these observations, the global distribution of errors does not correlate significantly with the seismic moment nor with the hypocentral depth or the focal mechanism of the earthquakes (Suppl. Figure 1).

For comparison, an empirical fit of $A_d$ against the seismic moment for the earthquakes in our database has an accuracy of and overall scatter similar to that of our model predictions (Figure 4, inset). Nevertheless, the fact that Eq 5, based on physical considerations and computed without any free parameter, has this accuracy for a global catalogue suggests that our approach captures essential aspects of earthquake-induced landsliding. Further validation of the model is inherently limited by the uncertainties associated with the estimation of $A_d$ and the inconsistency of the reported values for individual cases. However, additional insights into the validity and limitations of the model may be gained by comparing its prediction to objective landslide distribution areas obtained from well-constrained earthquakes. This is done in the next section for a limited number of comprehensive inventories where the landslide distribution is constrained in detail.

### 4.2 Model comparison against an objective measure of the landslide distribution area

To quantify the error of the model we evaluate the proportion of the total area affected by landsliding located within the $A_{dp}$ perimeter predicted by the model. We also consider the difference between the radius of the area affected by landsliding in the model, $R_{HMAX}$, and $R_{95}$ defined earlier as the distance from the seismic wave emission line within which 95% of the area affected by observed landsliding is located (Figure 2, 3).

For the ten earthquakes for which we have comprehensive landslide inventories, the model distribution area always contains



between 88% and 100% of the cumulative surface area of all mapped landslides (Table 2). These numbers indicate that the

model always captures the region of most intense landsliding, but that it sometimes overpredicts the affected area (when 100%

of the landslides are within the model distribution area) as for the Aysen, Niigata, and Iwate cases. The difference in 1D radius

gives a more accurate view of the merits and limits of the model (Table 2). For eight cases, the Northridge, Limon, Haiti, Aysen,

Finisterre, Guatemala, Wenchuan and Chi-Chi earthquakes, $R_{95}$ is well predicted, with an absolute error $< 6$ km, that is within

$\sim 20\%$ of $R_{95}$ in all cases (Figure 5). However, we note that the Limon inventory is incomplete in the most affected area,

suggesting that $R_{95}$ may actually be overestimated and $R_{HMAX}$ may excede it (Figure 3) Further, for the Haiti earthquake,

the model underpredicts $R_{95}$ by about 35% (9 km) if we consider Harp's 2016 inventory which extends far into little affected

areas (Figure 2). For the two remaining cases the error is much larger, with an over-prediction of about a factor 2, and an

absolute error of approximately 10 km, for the Niigata and Iwate earthquakes (Figure 3, 5). Additionally, we note that the

agreement between $R_{95}$ and our model may sometimes hide important along-strike variations or trends, as for the Guatemala

and Northridge earthquakes (Figure 2, 3). The possible reasons for such trends and other limitations of the model that could

explain why some cases are substantially under- or over-predicted are explored in the discussion, below.

We stress that the 5% of the total landsliding outside of $R_{95}$ may entail a significant hazard which can extend much further,

especially for large earthquakes like the Wenchuan, Chi-Chi or Guatemala cases which triggered huge amount of landslides

(total area of 1160, 128 and $61 km^2$, respectively). In the Wenchuan case, the distance from the emission line required to

encompass 97.5% of the total landslide area is of 48km instead of 34km for $R_{95}$. These landslides are more sparsely distributed

and most of the times smaller than the one close of the fault (Valagussa et al., In Review), but they remain difficult to predict

in many cases.

## 5 Discussion

We have shown that a first-order a-priori seismic shake map coupled with a universal shaking threshold for landsliding can

reproduce reasonably well the landslide distribution areas in a compilation of 83 cases, and that it matches the surface area

encompassing 95% of the total landslide area for most of the cases for which we have comprehensive landslide inventories. In

5 this section, we identify and try to quantify the different sources of uncertainties and potential ways to improve the model.

### 5.1 Asperity length and wave emission along the fault

For the Niigata, Iwate and Nagano earthquakes, the difference between the observed and predicted area of landsliding would

be greatly reduced if we would treat the earthquake as a single point source centered on the largest slip patch (Figure 3). This

illustrates the problem with the implicit assumption of Eq 5 that there is an equal emission of waves at the relevant frequencies

10 with source amplitude $b$, along the entire rupture. The amount of wave emission can vary along the rupture and may concentrate

in a zone much smaller than the rupture length. In the model of Marc et al. (2016b) the number of sources contributing to wave

emission and landsliding along a ruptured fault was given as $L/l_{asp}$, with $l_{asp}$ the characteristic length-scale of an asperity, set

arbitrarily to 3km. Thus, a 20 km long rupture is represented by six sources, distributed along most of the rupture length. The





Natural Hazards
and Earth System
efficacy of their model did not depend much on the value of $l_{asp}$ because their seismological model was calibrated empirically,
and $l_{asp}$ may well be larger. If we consider asperities as circular patches with diameter $\sim l_{asp}$, located strictly within the fault rupture and behaving as point sources with emission from their center, then the relevant length along which waves are emitted is $L - l_{asp}$ and Eq 5 becomes:

$$A_s = 2[L(Mo) - l_{asp}]sqrt(b(Mo)/a_c)^2 - R_0^2 H(L - l_{asp}) + pi(b(Mo)/a_c)^2 - R_0^2, \qquad (6)$$

with $H$ the Heaviside function, to represent that there should always be at least a point source in the middle of the fault even if it requires an asperity smaller than $l_{asp}$.

This has no significance for large earthquakes and long faults where $L >> l_{asp}$, but may significantly reduce the predicted value of $A_s$ for smaller earthquakes. This modification of the model, although plausible, would not improve much the global residuals because EQ 5 yields similar numbers of under-predicted and over-predicted small earthquakes, for which the model prediction is systematically reduced when using Eq 6. Moreover, cases such as the Niigata or Iwate earthquakes, are still over-predicted when modeled with a single point-source. This suggests that for these cases, with well-constrained source depth, a better prediction of $R_{HMAX}$ is needed, and therefore of either the source term $b_{sat}$, or the critical acceleration $a_c$.

### 5.2 Threshold acceleration for landsliding

Eq 2 can be rewritten as a second-degree equation and solved for $R_{HMAX}$ and $a_c$. Thus, assuming that Eq 4 and 3 hold and that $b_{sat} = 4000m$ is constant, we can use $A_d$, $C_{topo}$, $Mo$ and $R_0$ to invert for $R_{HMAX}$ and $a_c$ (Figure 6). Inverted values of $a_c$ cluster around $0.15$, that is the global value we have used here for the direct prediction, and about $50\%$ of the inverted values are between $0.1$ and $0.2$, consistent with values reported for case studies (Meunier et al., 2007; Hovius and Meunier, 2012; Yuan et al., 2013). The rest of the inverted values are mostly within the range $0.05 - 0.3$ and it is difficult to assess whether the inverted values of $a_c$ are representative of specifically weak or strong areas or whether other processes, not described by the model, affect the inversion. Defined as the threshold acceleration at which significant cohesion loss occurs, $a_c$ should be independent of hillslope steepness and depend only on material properties. Consistent with this view we find no correlation between $a_c$ and the landscape steepness as described by the modal slope of the landscape (cf., Marc et al., 2016b). To define and obtain quantitative estimates of substrate strength or of the ground pore pressure at the landscape scale is an outstanding challenge and lacking relevant constraints, we cannot assess further their influence on the variability of $a_c$ and $A_d$.

Nevertheless, it is interesting to focus on those earthquakes that have the lowest inverted values of $a_c < 0.05$. A striking example is the 1988 Saguenay (Canada) earthquake, which caused landsliding over an exceptionally large area ($A_d = 45,000km^2$), despite its moderate magnitude ($M_w = 5.8$) and large depth ($R_0 = 28km$). Eq 5 predicts no landsliding for this event, and the inverted value of $a_c$ is as low as $0.01$, more than an order of magnitude below the global threshold of $0.15$. The Virginia 2011 (USA) is a very similar example. One explanation is that the furthest landslides defining $A_d$ were indeed occuring at very low shaking levels (cf., Jibson and Harp, 2012) but seismological processes may also be in cause. The Saguenay earthquake was very peculiar from a seismological point of view, with a larger than expected stress drop (Boore and Atkinson, 1992), and therefore probably larger strong motions (e.g., Baltay et al., 2011; Baltay and Hanks, 2014). Also, its occurrence close to the



Moho discontinuity may have led to reflected waves reaching the surface with a stronger amplitude than the direct S-waves
(Somerville et al., 1990), causing strong shaking up to $\sim 100km$ from the epicenter. These effects have likely contributed to
an exceptional pattern of strong ground motion and a significantly extended landslide distribution area. Evidently, such effects
and mechanisms are not captured by our simple model. Therefore, inverting for $a_c$ and finding anomalous values, for example
$a_c < 0.05$, which is more than 3 times smaller than the global average, may suggest that an earthquake had an anomalous or
complex seismic process. Hence, low $a_c$ values suggest that the Bihar, and possibly the Whittier Narrow, Alaska and Wenchuan
earthquakes had some mechanistic specificities, either in wave emission or wave propagation, that have affected the distribution
of landsliding. For the Wenchuan case we have already highlighted the complex rupture and the importance of variable initial
stress state and rupture velocity (Wen et al., 2012a, b).

### 5.3 Source term and earthquake stress drop

The near-source wave amplitude, $b_{sat}$, is the only explicit parameter representing the earthquake source in our model. It was
kept constant in our analyses. However, as was the case in the Saguenay earthquake, various seismological processes may
increase or decrease the amplitude of waves emitted at the source of an earthquake. For example, it has been established that
earthquakes with larger dynamic stress drops must have stronger surface ground motions, especially at high frequency (Hanks
and McGuire, 1981; Baltay et al., 2011; Baltay and Hanks, 2014). For the 22 earthquakes with a constrained dynamic stress
drop (Table 1) we do not find a clear correlation between the residuals of our model and the magnitude of the stress drop (Suppl.
Figure 2). Except for three substantially under-predicted earthquakes (Saguenay, Arthur's Pass 1995, Erzincan) for which the
large $A_d$ may be due to relatively large stress drops, the 19 other residuals do not seem to be controlled by the stress drops.
Rupture speed and rupture speed variability may also influence the source emission and its characteristics (Wen et al., 2012a;
Causse and Song, 2015). It has been argued that rupture speed could correlate negatively with stress drop, blurring the relation
between earthquake stress drop and strong ground motion (Causse and Song, 2015). Accurate measurements of the rupture
speed and its variations are now made for large earthquakes (e.g., Wen et al., 2012a), but they are lacking for most historic
events or smaller events, impeding further exploration of their effects on landsliding.

### 5.4 Directivity and asymmetry in landslide distribution

We have shown that our model can predict the general distance from the emission line to a contour containing $95\%$ of all the
landslides as measured by their cumulative surface area, with a reasonnable reliability. However, this 1D parameter does not
describe potential 2D complexities in the shape of the landslide distribution area. For example, across-strike asymmetry often
exists for thrust faults where the hanging wall may experience larger shaking (Oglesby et al., 2000) and therefore has more
intense landsliding over a larger distance (Gorum et al., 2011). This hanging wall effect is difficult to isolate because on dip
slip faults the topography in the hanging wall is usually closer to the earthquake source and also more prone to landsliding due
to larger relief and steepness. Along-strike asymmetry of the landslide distribution may arise from asymmetry of the groun
shaking pattern due to seismic directivity. Directivity is the result of interference between waves emitted at different times and
locations along the rupture. It can enhance strongly the ground shaking in the direction of rupture propagation, but reduces





shaking at the fault tip opposite the rupture direction (Wallace and Lay, 1995). These effects were clearly articulated in the
1976 Guatemala earthquake, with both high seismic intensities and dense landsliding limited to a narrow band along the fault
near the rupture initiation, and spreading over a wider area at the other extremity of the fault (Harp et al., 1981) (Figure 3).

To further explore these effects, we attempt to quantify asymmetry of the landslide distribution and in the rupture mechanism
with very simple parameters available for most case. We define a landslide asymmetry indicator $L_{as} = 100*(A_{op}-A_{ep})/A_{tot}$,
with $A_{tot}$ the total landslide area and $A_{ep}$ and $A_{op}$ the total landslide area beyond a line perpendicular to the fault strike and
located at the fault tip closest to and furthest from the epicenter, respectively. This indicator effectively compares both fault
extremities and quantify the amount of the asymmetry in % of the total landsliding. This ratio is corrected for the relative
abundance of flat or gentle topography below the $10°$ threshold in the relevant areas. It can be compared to the distance between
the earthquake epicenter (projected on the fault trace) and the middle point of the fault rupture, normalized by the half length
of the fault rupture, $Epi_{as}$. This latter measure tends to zero for a centered epicenter and to one for an epicenter at the tip of the
fault, suggesting low and high directivity, respectively. These indices cannot be computed for the Finisterre, Aysen and Limon
earthquakes for which we do not have constraints on the location of the actual portion of the fault that ruptured and its position
relative to the epicenter. For the other cases, we find a degree of correlation between the asymmetry of the earthquake epicenter
and the landslide distribution, at least where $Epi_{as} > 0.4$, but results for the 10 cases with comprehensive landslide inventories
are not straightforward (Figure 7).

For $Epi_{as} > 0.4$ we observen negligible landslide asymmetry. For larger $Epi_{as}$ values we observe a strong asymmetry oriented
with the directivity for the Northridge and Guatemala earthquakes, a moderate asymmetry for the Iwate case. For the Wenchuan
case, we find little (10%) asymmetry in the landslide distribution, even though the earthquake hypocenter was at one extremity
of the fault rupture. In this earthquake, the rupture propagated over multiple fault segments with different initial stresses, with
important rupture speed changes at the transition between segments (Wen et al., 2012a). This is likely to have complicated the
ground shaking pattern (Wen et al., 2012b; Meunier et al., 2013), blurring any directivity effect. Similar effects may be at play
for the Gorkha and Denali earthquakes where published landslide maps (Gorum et al., 2014; Martha et al., 2016) indicate little
(10%) to no asymmetry ($< 1\%$), respectively (Figure 7). These cases also had complex ruptures, with the Denali earthquake
rupturing three segment, the last one in super-shear (Frankel, 2004), and the Gorkha earthquake propagating along a complex
fault geometry with laterally varying mechanism (Kumar et al., 2016). Finally, we note that the amplitude of ground shaking
is reduced by directivity but the shaking duration is longer, possibly balancing any effects on landslide triggering. In summary,
the landslide asymmetry proxy proposed here varies between earthquakes and is not simply related to the epicenter position
relative to the fault trace. Analysis of a larger number of well-constrained cases is necessary to constrain and quantify the effect
of seismic directivity on landslide patterns.

## 6   Conclusions

We have presented an analytical expression for the distribution area of earthquake-induced landslides. It shares its derivation
with the model of Marc et al. (2016b) that has been shown to predict total landslide volume and area with good accuracy in





most cases. The expression is based on scaling relationships between essential seismic parameters such as the seismic moment,
15  the source depth, the rupture length and aims at predicting where the shaking level is likely to exceed a universal threshold
for rock damage and onset of landsliding. Compared to a large compilation of estimates of landslide distribution areas from
observational constraints, the analytical expression is shown to explain 56% of the variance without any adjusted parameters,
and to predict 35 or 49 of 83 cases within a factor of 2, when taking the hypocentral depth as the emission depth, or when
allowing the emission depth to be within 25% of the hypocentral depth, respectively. Analysis of outliers is not easy with such
compilations as many cases are poorly constrained and because the definition and measurement of the landslide distribution
area are not uniform for all cases. For detailed inventories we see that the prediction of our model agrees relatively well with
the region in which 95% of the total landslide area is concentrated. However, some earthquakes are significantly overpredicted
and others have important along-strike asymmetry not captured by our model. These discrepancies may arise from variability
in wave emission along a complex rupture (e.g., Wen et al., 2012a), as well as wave interference leading to directivity. Thus,
modelling of the ground shaking pattern must be improved to aid better understanding and forecasting of landslide hazard
associated with earthquakes. Nevertheless, the overall agreement of our prediction is striking given that it does not use any
geomorphological calibration. It suggests that, although important departures from the universal acceleration threshold of
0.1 to 0.2 g (Meunier et al., 2007; Marc et al., 2016b) exist locally (e.g., Jibson and Harp, 2016), they may be of minor
significance to the bulk behaviour of earthquake-triggered landsliding. Still, understanding how critical acceleration varies
from a landscape to another is probably key to increase the accuracy of our prediction as much as better understanding control
on the ground shaking. The prediction of landslide distribution area, together with the recent expressions for total volume
and area of landslides (Marc et al., 2016b) and constraints on the spatial pattern of landslide density and size (Meunier et al.,
2007; Valagussa et al., In Review), defines a consistent framework for the evaluation of seismological parameters controlling
ground shaking for quantitative landslide hazard. Finally, we note that the simplicity and limited number of parameters of
the expression presented here makes it well-suited for hazard assessment in earthquake scenarios as well as in the immediate
aftermath of an earthquake.

## 7 Data availability

The data used in this study is available through the data table or through the literature.

*Author contributions.*  OM collected and analyzed all data. OM conceived the study and wrote the paper with assistance and inputs from NH
and PM.

*Competing interests.*  The author declare no conflicts of interest with the work presented in this study.



*Acknowledgements.*  The authors are grateful to Xu C. and Gorum T. for providing their landslide inventories. The authors have gratefully used Aster GDEM V2, a product of METI and NASA.

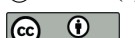



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





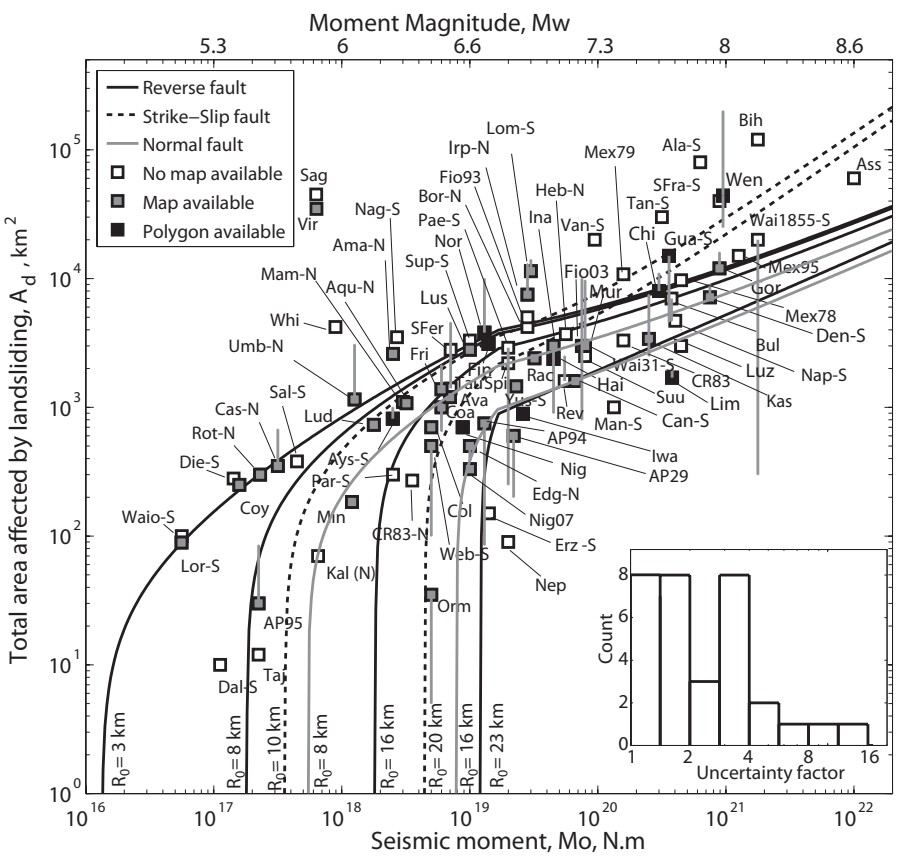

**Figure 1.** Total area affected by landsliding against seismic moment and moment magnitude for 83 earthquakes. Vertical errors bars represent different estimate of $A_d$ for the 27 cases where they could be obtained. Name code (defined in Table 1) are shown for each earthquakes, followed by S or N for strike-slip and normal faults, while all other cases are reverse fault earthquakes. The prediction of the seismologically consistent model is shown for reverse, normal and strike-slip faults at different depth with solid black, solid grey and dashed lines, respectively. Inset: Distribution of the uncertainty factor ( Upper Estimate / Best estimate or Best Estimate / Lower Estimate).




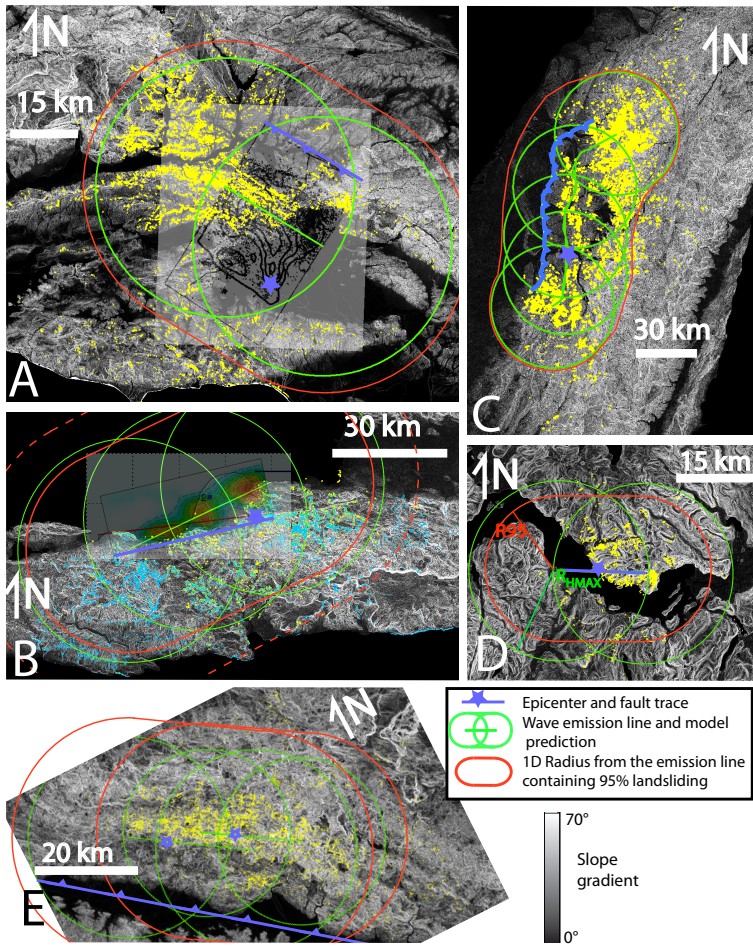

**Figure 2.** Distribution of landsliding (yellow polygons) and predicted landslide distribution area (green circles) for the Northridge (A), Haiti (B), Chi-Chi (C), Aysen (D) and Finisterre (E) earthquakes. The background is a topographic slope gradient map derived from 30m-Aster GDEM, provided by NASA, allowing to locate flatlands where no landslides are expected even if the shaking threshold is exceeded. Emission line source and the area where the ground shaking is expected to be larger than $a_c$ are represented with green lines and circles, respectively. Red perimenters show the area encompassing 95% of the total area of landsliding defined by a uniform distance away from the wave emission line. For reference rupture slip distribution maps are shown for the Northridge and Haiti earthquakes (Wald et al., 1996; Hayes et al., 2010). For Haiti, in addition to the landslide inventory from Gorum et al. (2013), we also show the one of Harp et al. (2016) (cyan polygons) and its associated $R_{95}$ with red dashed line. Note that the Aysen strkie-slip fault is located only based on the focal mechanism and epicenter, and that the other solution (north oriented) would give a similar radius for $R_{95}$.





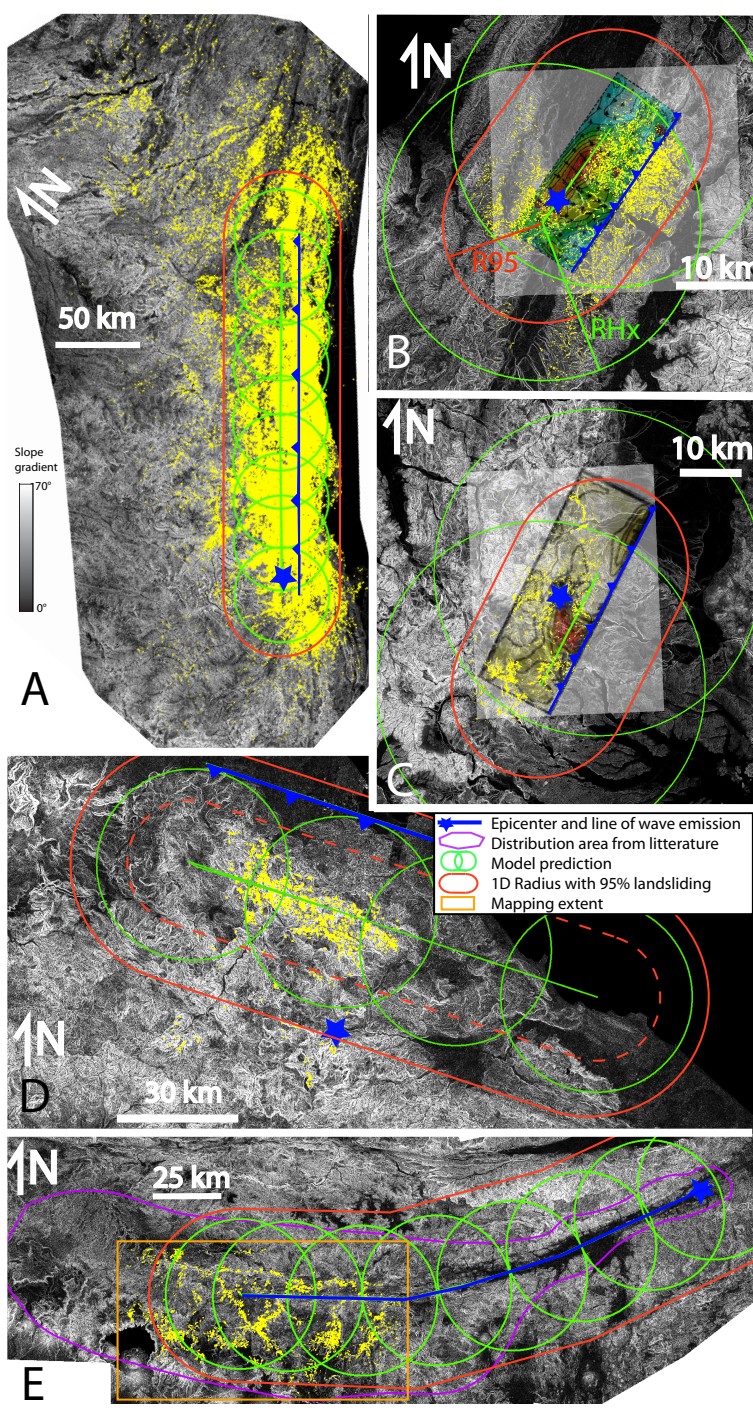

**Figure 3.** Same as Figure 2 for the Wenchuan (A), Niigata (B), Iwate (C), Limon (D) and Guatemala (E) earthquakes. For reference rupture slip distribution maps are shown for the Niigata and Iwate earthquakes (Hikima and Koketsu, 2005; Suzuki et al., 2010). Additionally, the distribution area proposed by Harp et al. (1981) for the Guatemala earthquake is shown in purple. For the Limon earthquake, the total area of landslides is likely underestimated in the most affected area and therefore $R_{95}$ is likely reduced and more similar to the dashed red contour.





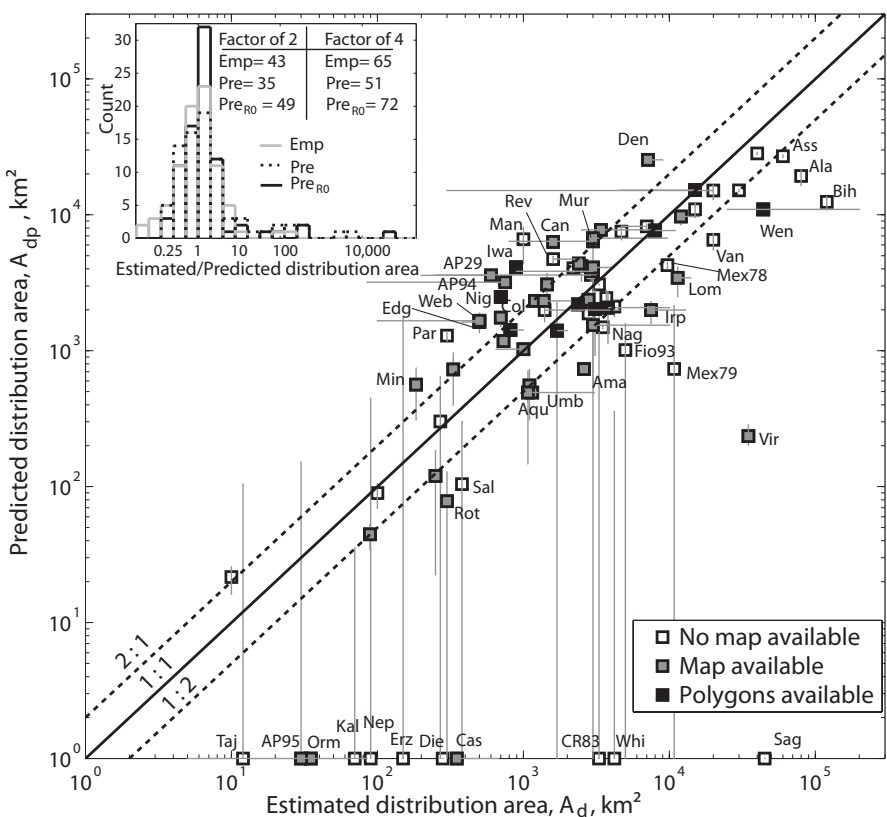

**Figure 4.** Predicted landslide distribution area plotted against estimated landslide distribution areas for 83 earthquakes. For visibility, cases where the predicted area is 0 are set to $1km^2$ and only the name codes of earthquakes outside of a factor of 2 from the 1:1 line are shown. Vertical error bars represent the range of predicted values when $R_0$ is varied between 75 and 125% of the best estimate of the hypocentral depth. Inset: Histograms of model residuals ($A_d$ / $A_{dp}$). Histogram for the best empirical fit of $A_d$ against $Mo$ (Emp, grey line), for the model prediction (Pre, dashed line) and for the prediction accounting for 25% of uncertainty on $R_0$ ($Pre_{R0}$, solid black line) are shown, with the number of earthquakes correctly predicted within a factor of 2 and 4.





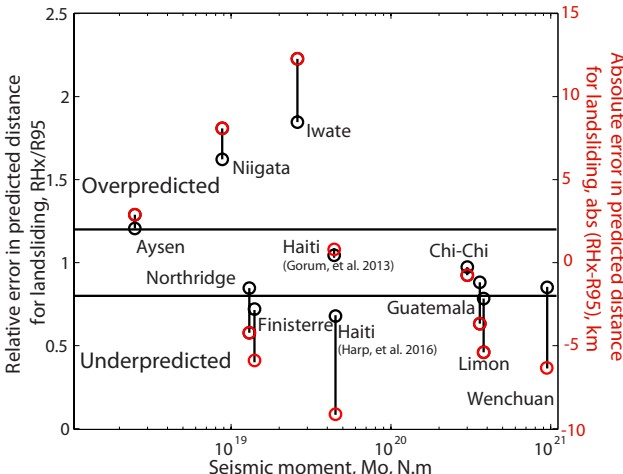

**Figure 5.** Relative and absolute errors in the prediction of the distance from the wave emission line containing 95% of the total landslide area plotted against the seismic moment. Horizontal black lines delimit cases where the relative error is within $< 20\%$

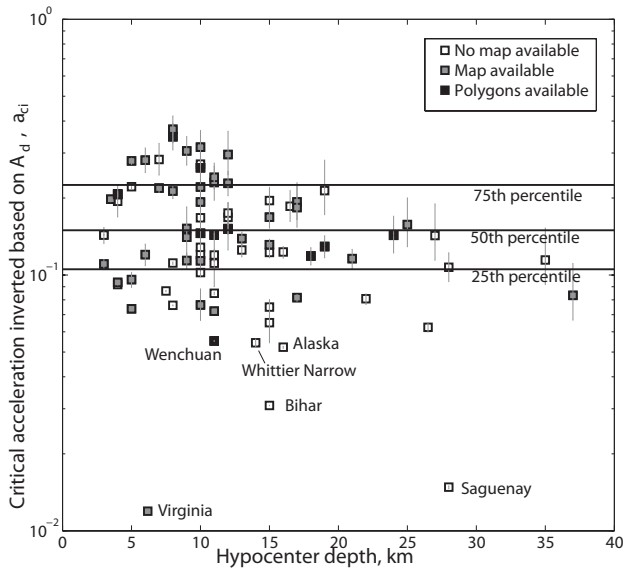

**Figure 6.** Inverted value of the critical acceleration, $a_c$, against hypocentral depth. Most values cluster between 0.1 and 0.2, but some events (with their neame tags displayed) have exceptionally low inverted values cases.





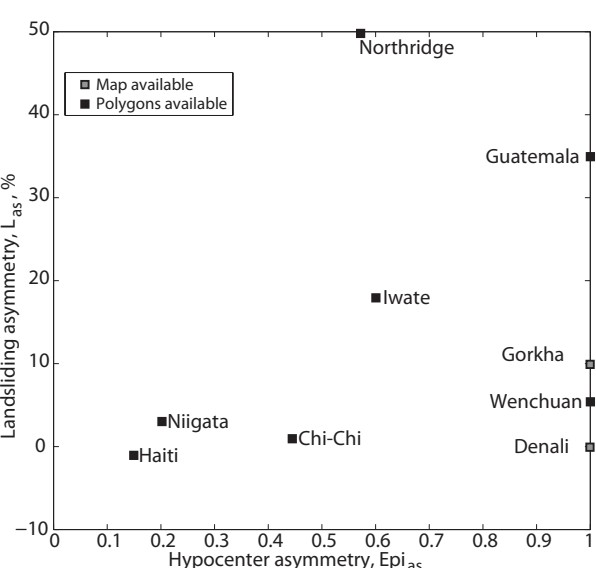

**Figure 7.** Hypocenter asymmetry (0= center, 1= tip of the fault) against along-strike landsliding asymmetry, defined by the difference between the total landslide area beyond the tip of the fault opposed to the epicenter and the total landslide area beyond the other tip normalized by the total landslide area.





**Table 1.** Data summary. Earthquake contains location and country where US, NZ, CN, and IT are for United States, New Zealand, China, and Italy. Fault gives the focal mechanism (Reverse, Strike-slip or Normal). Depth is the hypocenter depth. The best estimate of $A_d$ is followed by lower and upper bound into brackets. Q indicates if a map or an inventory with polygons could be accessed. $\Delta\sigma$ refers to the dynamic stress drop. Numbers in the references are as follow: 1= (Keefer, 1984); 2= (Hancox et al., 1997); 3= (Rodriguez et al., 1999); 4= (Bommer and Rodriguez, 2002); 5= (Martino et al., 2014); 6= (Mosquera-Machado et al., 2009); 7= (Kamp et al., 2008); 8= (Gorum et al., 2014)

| Earthquake | Code | Year | Fault | $M_w$ | Depth (km) | $A_d$ ($km^2$) | Q | $C_{topo}$ | $\Delta\sigma$ (MPa) | Ref. |
|---|---|---|---|---|---|---|---|---|---|---|
| Wairapa (NZ) | Wai1855 | 1855 | SS | 8.10 | 16 | 20000 [300,-] | na | 0.4 | na | 2 |
| Canterbury (NZ) | Can | 1888 | SS | 7.15 | 10 | 1600 [-,-] | Map | 1 | na | 2 |
| San Francisco (US) | SFra | 1906 | SS | 7.90 | 8 | 40000 [-,-] | na | 1 | na | 1 |
| Bueller (NZ) | Bul | 1929 | R | 7.65 | 10 | 7000 [4700,-] | na | 1 | na | 2 |
| Arthur's Pass (NZ) | AP29 | 1929 | R | 6.83 | 12 | 600 [200,-] | Map | 1 | na | 2 |
| Napier (NZ) | Nap | 1930 | SS | 7.67 | 15 | 4700 [4700,-] | na | 0.5 | na | 2 |
| Wairoa (NZ) | Wai31 | 1931 | SS | 7.18 | 12 | 3000 [800,-] | Map | 1 | na | 2 |
| Bihar (Nepal) | Bih | 1934 | R | 8.10 | 15 | 120000 [-,-] | na | 1 | na | 1 |
| Vancouver (Canada) | Van | 1946 | SS | 7.25 | 15 | 20000 [-,-] | na | 1 | na | 1 |
| Coleridge Lake (NZ) | Col | 1946 | R | 6.40 | 10 | 700 [-,-] | Map | 1 | na | 2 |
| Assam (India) | Ass | 1950 | R | 8.60 | 8 | 60000 [-,-] | na | 1 | na | 1 |
| Daly City (US) | Dal | 1953 | SS | 5.30 | 4 | 10 [-,-] | na | 0.15 | na | 1 |
| Alaska (US) | Ala | 1958 | SS | 7.80 | 16 | 80000 [-,-] | na | 1 | na | 1 |
| Hebgen Lake (US) | Heb | 1959 | N | 7.10 | 11 | 3700 [-,-] | na | 1 | na | 1 |
| Parkfield (US) | Par | 1966 | SS | 6.20 | 7 | 300 [-,-] | na | 1 | na | 1 |
| Inangahua (NZ) | Ina | 1968 | R | 7.04 | 15 | 3000 [900,-] | Map | 1 | na | 2 |
| San Fernando (US) | SFer | 1971 | R | 6.50 | 13 | 2800 [-,-] | na | 1 | na | 1 |
| Tangshan (CN) | Tan | 1976 | SS | 7.60 | 7.5 | 30000 [-,-] | na | 1 | na | 1 |
| Friuli (IT) | Fri | 1976 | R | 6.45 | 4 | 1380 [-,2125] | Map | 1 | na | 5 |
| Guatemala | Gua | 1976 | SS | 7.64 | 12 | 15000 [4600,-] | Poly | 1 | na | 1, Harp et al., 1981 |
| Mexico | Mex78 | 1978 | R | 7.70 | 11 | 9700 [-,-] | na | 0.5 | na | 4 |
| Mexico | Mex79 | 1979 | R | 7.40 | 26.5 | 10800 [-,-] | na | 0.5 | na | 4 |
| Coyote Lake (US) | Coy | 1979 | SS | 5.40 | 6 | 250 [-,-] | Map | 1 | na | 1, Keefer et al., 1980 |
| Mammoth Lake (US) | Mam | 1980 | N | 6.25 | 9 | 1100 [-,-] | Map | 1 | 0.276 | 1, Harp et al., 1984 |
| Irpina (IT) | Irp | 1980 | N | 6.90 | 11 | 7500 [-,13000] | Map | 1 | na | 3,5 |
| Costa Rica | CR83 | 1983 | R | 7.40 | 28 | 3300 [-,-] | na | 0.5 | na | 4 |
| Costa Rica | CR83 | 1983 | N | 6.30 | 12 | 270 [-,-] | na | 1 | na | 4 |
| Borah Peak (US) | Bor | 1983 | N | 6.90 | 10 | 4200 [-,-] | na | 1 | 4.048 | 3 |
| Waiotapu (NZ) | Waio | 1983 | SS | 5.10 | 3 | 100 [-,-] | na | 1 | na | 2 |
| Coalinga (US) | Coa | 1983 | R | 6.45 | 9 | 1000 [650,-] | Map | 0.5 | 0.506 | 3, Harp et al., 1990 |
| Nagano (Japan) | Nag | 1984 | SS | 6.22 | 4 | 3500 [-,-] | na | 1 | na | 3 |
| Kalamata (Greece) | Kal | 1986 | N | 5.81 | 11 | 70 [-,-] | na | 1 | na | 3 |
| Salvador | Sal | 1986 | SS | 5.70 | 10 | 380 [-,-] | na | 1 | na | 3 |
| Diebu (CN) | Die | 1987 | R | 5.37 | 15 | 280 [-,-] | na | 1 | na | 3 |
| Whittier Narrow (US) | Whi | 1987 | R | 5.90 | 14 | 4200 [-,-] | na | 1 | na | 3 |
| Superstition Hills (US) | Sup | 1987 | SS | 6.60 | 3 | 3300 [-,-] | na | 1 | na | 3 |
| Reventador (Ecuador) | Rev | 1987 | R | 7.09 | 10 | 1600 [-,2500] | Map | 1 | na | 3, Tibaldi et al., 1995 |
| Edgecumbe (NZ) | Edg | 1987 | N | 6.60 | 6 | 500 [380,-] | Map | 1 | na | 2 |
| Nepal | Nep | 1988 | R | 6.80 | 35 | 90 [-,-] | na | 1 | na | 3 |



| | | | | | | | | | |
|---|---|---|---|---|---|---|---|---|---|
| Saguenay (Canada) | Sag | 1988 | R | 5.80 | 28 | 45000 [-,-] | na | 1 | 11 | 3 |
| Spitak (Armenia) | Spi | 1988 | R | 6.80 | 5 | 2200 [-,-] | na | 1 | na | 3 |
| Tajikistan | Taj | 1989 | R | 5.50 | 10 | 12 [-,-] | na | 1 | na | 3 |
| Loma Prieta (US) | Lom | 1989 | SS | 6.92 | 17 | 11400 [-,14000] | Map | 1 | 6.348 | 3, Keefer and Manson, 1998 |
| Manyil (Iran) | Man | 1990 | SS | 7.35 | 19 | 1000 [-,-] | na | 1 | na | 3 |
| Weber (NZ) | Web | 1990 | SS | 6.40 | 11 | 500 [100,-] | Map | 1 | na | 2 |
| Luzon (Philippines) | Luz | 1990 | SS | 7.70 | 25 | 3000 [-, -] | na | 0.5 | na | 3 |
| Racha (Georgia) | Rac | 1991 | R | 6.94 | 7 | 2400 [-,-] | Map | 1 | 1.4 | Jibson et al., 1994 |
| Limon (Costa Rica) | Lim | 1991 | R | 7.65 | 24 | 1700 [-,2000] | Poly | 0.35 | 0.064 | 3, T |
| Erzican (Turkey) | Erz | 1992 | SS | 6.70 | 27 | 150 [-,-] | na | 1 | 2.5 | 3 |
| Suusanmyr (Kyrgyzstan) | Suu | 1992 | R | 7.20 | 16.5 | 2500 [-,-] | na | 1 | 0.7 | 3 |
| Fiorland (NZ) | Fio93 | 1993 | R | 6.90 | 22 | 5000 [-,-] | na | 0.5 | na | 2 |
| Ormond (NZ) | Orm | 1993 | R | 6.40 | 37 | 35 [5,-] | Map | 1 | na | 2, 3 |
| Finisterre (New Guinea) | Fin | 1993 | R | 6.70 | 19 | 3100 [-,-] | Poly | 1 | 1.9 | Meunier et al., 2008 |
| Paez (Colombia) | Pae | 1994 | SS | 6.80 | 12 | 2900 [250,-] | na | 1 | na | 3 |
| Arthur's Pass (NZ) | AP94 | 1994 | R | 6.68 | 9 | 750 [85,-] | Map | 1 | 0.3 | 2,3 |
| Northridge (US) | Nor | 1994 | R | 6.68 | 18 | 3800 [-,10000] | Poly | 1 | 5.428 | 3, T |
| Mexico | Mex95 | 1995 | R | 8.00 | 15 | 15000 [-,-] | na | 1 | na | 4 |
| Tauranema (Colombia) | Tau | 1995 | R | 6.50 | 12 | 1400 [-,4550] | na | 1 | 0.2 | 3, 4 |
| Arthur's Pass (NZ) | AP95 | 1995 | R | 5.50 | 9 | 30 [-, 85] | Map | 1 | 1.3 | 2, 3 |
| Murindo (Colombia) | Mur | 1995 | SS | 7.20 | 11 | 3000 [-, 9700] | Map | 1 | 0.3 | 4, 6 |
| Umbria-Marche (IT) | Umb | 1997 | N | 6.00 | 4 | 1150 [-, 3075] | Map | 1 | na | 5 |
| Castellucio (IT) | Cas | 1998 | N | 5.60 | 10 | 350 [-, 675] | Map | 1 | na | 5 |
| Chi-Chi (Taiwan) | Chi | 1999 | R | 7.58 | 10 | 8000 [-,11000] | Poly | 1 | 0.4 | Liao and Lee, 2001 |
| Avaj (Iran) | Ava | 2002 | R | 6.50 | 8 | 1200 [-,-] | Map | 1 | 0.4 | Madhavifar et al., 2006 |
| Denali (US) | Den | 2002 | SS | 7.85 | 8 | 7150 [-,9000] | Map | 1 | 0.6 | Gorum et al., 2014 |
| Fiorland (NZ) | Fio03 | 2003 | R | 7.18 | 21 | 3000 [-,10000] | Map | 0.5 | 0.11 | Hancox et al., 2003 |
| Rotoehu (NZ) | Rot | 2004 | N | 5.51 | 5 | 300 [-,-] | Map | 1 | na | Hancox et al., 2004 |
| Niigata (Japan) | Nig | 2004 | R | 6.56 | 10 | 700 [-,-] | Poly | 1 | 0.9 | Yagi et al., 2007 |
| Kashmir (Pakistan) | Kas | 2005 | R | 7.53 | 5 | 3400 [2500, 7500] | Map | 1 | 1.5 | Sato et al., 2007, 7 |
| Niigata (Japan) | Nig07 | 2007 | R | 6.60 | 17 | 331 [-,-] | Map | 0.4 | na | Collins et al., 2012 |
| Aysen Fjord (Chile) | Ays | 2007 | SS | 6.20 | 4 | 815 [-,1000] | Poly | 1 | na | 8 |
| Iwate (Japan) | Iwa | 2008 | R | 6.88 | 8 | 890 [-,-] | Poly | 1 | 2.3 | Yagi et al., 2009 |
| Wenchuan (CN) | Wen | 2008 | R | 7.92 | 11 | 44000 [-,200000] | Poly | 1 | na | Xu et al., 2014 |
| Aquila (IT) | Aqu | 2009 | N | 6.27 | 10 | 1075 [-,-] | Map | 1 | na | 5 |
| Yushu (CN) | Yus | 2010 | SS | 6.84 | 17 | 1455 [-,-] | Map | 1 | na | Xu et al., 2014 |
| Lorca (Spain) | Lor | 2010 | SS | 5.10 | 3 | 89 [-,-] | Map | 0.5 | na | Alfaro et al., 2011 |
| Haiti | Hai | 2010 | R | 7.04 | 11 | 2300 [-,3800] | Poly | 0.5 | na | 8, Harp et al., 2016 |
| Virginia (US) | Vir | 2011 | R | 5.80 | 6 | 33400 [-,-] | Map | 0.5 | na | Jibson et al., 2012 |
| Lushan (CN) | Lus | 2013 | R | 6.60 | 13 | 2800 [-,-] | Map | 1 | na | Xu et al., 2015 |
| Minxian (CN) | Min | 2013 | R | 5.99 | 10 | 184 [-,-] | Map | 1 | na | Xu et al., 2014 |
| Ludian (CN) | Lud | 2014 | R | 6.10 | 3.5 | 731 [-,-] | Map | 1 | na | Zhou et al., 2016 |
| Gorkha (Nepal) | Gor | 2015 | R | 7.90 | 15 | 12000 [-,16000] | Map | 1 | na | Martha et al., 2016 |
| Amatrice (IT) | Ama | 2016 | N | 6.20 | 5 | 2600 [-,-] | Map | 1 | na | 5 |





**Table 2.** Summary of the emission length, $L$, the maximum horizontal radius for landsliding, $R_{HMAX}$, the landsliding included in the prediction (% of $A_{tot}$), and the distance from the emission line containing 95% of the total landslide area, $R_{95}$. In parenthesis we give the values of $L$ derived from rupture length scaling with moment (EQ 3) that were used when we could not access a rupture slip model. Note that for the Haiti earthquake we show the results from the inventory of Gorum et al. (2013) and of Harp et al. (2016) (in bracket). Note that we use the two largest shocks to model the Finisterre event. Last, note that for the Limon earthquake some landslides in the most affected area could not be delineated and the total area is underestimated. Therefore we likely underestimate the amount of landsliding within the model prediction and overestimate $R_{95}$, that may actually be smaller than $R_{HMAX}$.

| EQ | Northridge | Niigata | Iwate | Finisterre | Wenchuan | Chi-Chi | Haiti | Aysen | Guatemala | Limon |
|---|---|---|---|---|---|---|---|---|---|---|
| $L$, km | 21 (30) | 15 (26) | 20 (39) | (25/20) | 220 (170) | 90 (106) | 40 (49) | (15) | 180 (260) | (115) |
| $R_{HMAX}, km$ | 23.5 | 21 | 27 | 25/20 | 28 | 27 | 26 | 17 | 27 | 20 |
| $\% of A_{tot}$ | 90 | 99 | 100 | 98 | 91 | 94 | 96[88] | 98 | 88 | 93* |
| $R_{95}, km$ | 27.5 | 13 | 14.5 | 31/26 | 34 | 28 | 25[35] | 14 | 31 | 25* |