# Peer review of "Prediction of the area affected by earthquake-induced landsliding based on seismological parameters"

_Natural Hazards and Earth System Sciences, 2017_

## Referee Comment (RC1) · Anonymous Referee #1 · 2 Mar 2017

GENERAL COMMENT:

This is a well-written paper that does a very good job of providing a seismologically consistent relation between seismic energy release and the area affected by triggered landslides. The data sets used are appropriate and well documented, and the model is explained clearly even if some aspects of it are somewhat complex. The principal limitation of the paper is that it takes an almost purely seismological view of the problem and minimizes or dismisses geomorphic and geotechnical considerations, which, in fact, play a key role in the susceptibility to landsliding and thus the ultimate landslide distribution. By better addressing these concerns and taking into account both material-based and topographic effects on critical acceleration, the paper can be strengthened considerably and be a valuable contribution to the literature aimed at characterizing seismic landslide hazards.

SPECIFIC COMMENTS: [Note: line numbering in the draft manuscript is inconsistent and non-unique; this is a best attempt to identify the locations of technical comments]

Page 2, paragraph 1: Examining the zone of concentrated landsliding rather than the extreme limits of landsliding is a sound approach to eliminate outliers and unusual conditions.

Page 2, paragraph 2: Suggest adding Keefer (2002) to this list of references. His updated paper contains additional data.

Page 2, paragraph 2: This relation is parallel to the relation between Arias intensity and seismic moment developed by Wilson and Keefer (1985, p. 334). This is an early and somewhat archaic reference, but it laid the groundwork for the kind of modeling done in the current paper and probably should referenced.

Page 3, paragraph 2: Unclear what the term "oversteepened slopes" means here. "Oversteepened" generally means that some geomorphic process has created a slope having marginal static stability; active cutbanks of rivers are an example this. But earth-quakes trigger landslides on slopes that are perfectly stable in static conditions but that fail under seismic loading. It is not a matter of oversteepening. And the next line states that critical acceleration (ac) is independent of slope angle, so why would only over-steepened slopes be more susceptible to failure? These statements are inconsistent.

Page 3, paragraph 2: Critical acceleration needs to be expressed in terms of units. Presumably this is in terms of g, the acceleration of gravity (this is an issue throughout the paper). The selection of ac=0.15 g seems quite arbitrary and needs justification. Failure of slopes having ac=0.05 g or less is not uncommon. Provide justification.

Page 3, paragraph 2: The statement that critical acceleration is independent of hillslope gradient is completely false. In fact, critical acceleration is exquisitely sensitive to slope steepness. The basic equation defining critical acceleration [ac=(FS-1)g sin(a), where FS is static factor of safety and a is slope angle] contains the slope gradient both

explicitly in the equation and implicitly in the FS term. Jibson et al. (2000) showed that critical acceleration is strongly dependent on slope angle and less so on modest variations in material strength. That is the exact opposite of what is stated here. How is this contradiction justified?

Page 4, paragraph 2: The suggestion that critical acceleration is spatially constant is very difficult to justify. Jibson et al. (2000) show clearly that landslide distribution is a function of the interaction of the spatial variation in ground shaking and the spatial variation in critical acceleration. Treating the latter as if it were constant and simply assuming that the landslide distribution is purely a function of ground-motion variation is naïve and physically unrealistic. Critical acceleration typically varies wildly over an area shaken by an earthquake. Without relitigating Marc et al. (2016) it is necessary to provide some basis for this far-reaching and questionable assumption.

Page 7, paragraphs 2 and 3: This is a good way to define the area affected by landslides that eliminates outliers on slopes having anomalously low critical accelerations. And this should encourage more polygon inventories in the future, which are becoming the norm.

Page 9, paragraph 1: Why not use a finite-fault model and examine distances from the point or area of maximum moment release?

Page 10, paragraph 3: Critical acceleration is not defined as loss of cohesion. Cohesion is only one component of shear strength. Critical acceleration is the acceleration necessary to overcome the resisting shear strength (both frictional and cohesive) and initiate permanent landslide displacement.

Page 10, paragraph 3: Critical acceleration is in no way independent of hillslope steepness. It is, in fact, extremely sensitive to slope. Not sure where this idea came from, but it is demonstrably, mathematically false. As indicated above, slope enters into the critical acceleration equation in two different places and thus is doubly important.

Page 10, paragraph 3: The statement that better characterization of strength and pore pressure is necessary to refine estimates of critical acceleration is an understatement. Dreyfus et al. (2013) discussed this and should be cited here.

Page 10, paragraph 3: The Saguenay earthquake is anomalous not just seismologically but also in terms of the types of landslides triggered. Ground-motion amplitude was moderate, but the duration and thus number of cycles of shear stress was fairly high. The landslides triggered by the earthquake included sensitive-clay failures and slides related to liquefaction, which are more sensitive to duration than amplitude of ground shaking. The type of landslide matters as much as the peculiar seismology. Another way of saying this is that different characteristics of ground shaking (amplitude, frequency, duration) will trigger different types of landslides, and susceptibility to those types of landslides will determine the extent of landsliding. It is more than just seismology.

Page 11, paragraph 3: Here it states that slope steepness makes an area more susceptible to landsliding, but on the previous page it is stated that critical acceleration (the measure of seismic landslide susceptibility) is independent of slope steepness. This is strongly inconsistent.

Page 13, paragraph 1: The range of 0.1-0.2 g is not accepted as a "universal acceleration threshold." The Jibson and Harp (2016) study of several of the best documented earthquakes (in terms of landslides) suggests a threshold closer to 0.05 g. The difference is between the outermost limit of the smallest landslides and the zone of concentrated landsliding. This differentiation should be made clearer here. The threshold acceleration values in the different studies are really looking at different landslide limits.

Figure 1: Define Ro in caption.

Figure 5: Not clear what the red circles indicate.

Figure 6: Typo in caption: "name."

References:

Jibson, R.W., Harp, E.L., and Michael, J.A., 2000, A method for producing digital probabilistic seismic landslide hazard maps: Engineering Geology, v. 58, p. 271-289.

Wilson, R.C., and Keefer, D.K., 1985, Predicting areal limits of earthquake-induced landsliding, in Evaluating Earthquake Hazards in the Los Angeles Region—An Earth-Science Perspective, J.I. Ziony (Editor): U.S. Geological Survey Professional Paper 1360, p. 317-345.

---

## Author Comment (AC1) · 6 Mar 2017

We are pleased by the encouraging and clear review done by referee 1. In this preliminary reply we mainly aim at clarifying the meaning of $a_c$ and why we define it like this. We believe this will resolve most of the recurring problem the referee have underlined with $a_c$ in his comments. We hope the added sentences (Page C3 of this comment) make the definition of $a_c$ clear and sound and encourage both referee to double check they are sufficient.

The Line numbering is following the NHESSD Latex Template, with a Page / Line schemes. We will use it to locate changes related to specific comments. Below the comments are repeated and addressed.

Page 2, paragraph 1: Examining the zone of concentrated landsliding rather than the

[Figure]

extreme limits of landsliding is a sound approach to eliminate outliers and unusual conditions.

» Agreed. We will add a similar comment on Page 2 Line 6/7

Page 2, paragraph 2: Suggest adding Keefer (2002) to this list of references. His updated paper contains additional data.

» Page 2 Line 12: We will include Keefer 2002.

Page 2, paragraph 2: This relation is parallel to the relation between Arias intensity and seismic moment developed by Wilson and Keefer (1985, p. 334). This is an early and somewhat archaic reference, but it laid the groundwork for the kind of modeling done in the current paper and probably should referenced.

» Page 2 Line 31: We will also include Wilson and Keefer 1985

Page 3, paragraph 2: Unclear what the term "oversteepened slopes" means here. "Oversteepened" generally means that some geomorphic process has created a slope having marginal static stability; active cutbanks of rivers are an example this. But earthquakes trigger landslides on slopes that are perfectly stable in static conditions but that fail under seismic loading. It is not a matter of oversteepening. And the next line states that critical acceleration (ac) is independent of slope angle, so why would only oversteepened slopes be more susceptible to failure? These statements are inconsistent.

» Page 3 Line 8: By oversteepened we mean here slopes that are steeper than the coefficient of friction of the material and therefore stable only because of a relatively high cohesion. We do not want to refer to a specific mechanism, and oversteepened is probably misleading.

Our hypothesis is that these "Cohesion Stabilized slopes" will fail if the repeated cycles of strong motion cause a sufficient drop of cohesion. Therefore $a\_c$ is not the critical acceleration defined compare to a static force balance and factor of safety, but we define it as a material property, setting the acceleration at which damage (cohesion

reduction) initiate. This was more developed in Marc et al 2016b, and we now have developed it here too, to avoid confusion.

We will replace: "Over-steepened" slopes by " slopes above their friction angle and thus stable due to a significant cohesion".

Then we will add: "Therefore in this paper, a_c is not related to the safety factor and slope gradient of a given hillslopes, but only defines at which level of acceleration damage and cohesion reduction will initiate (cf. Marc et al., 2016b for more details). With this definition we consider that a_c must vary modestly compared to cohesion that can vary greatly between soil and fractured or fresh rock. This is consistent with modest variations of the minimum acceleration at which landslide occurred estimated for landslide mapped by satellite (0.1-0.2g Meunier 2007, Hovius and Meunier 2012, Yuan 2013) or minor rockfall (Jibson and Harp 2016). The initiation of soil non linear behavior has also been observed around 0.15g (Wen 1994). The average value of a_c across a landscape will be important to define A_d, and we initially assume that mean (a_c)∼ 0.15g to be conservative and focus on the area where significant landsliding occur. Note, that in this approach, a_c is independent of the slope gradient, but also that a slope experiencing an acceleration larger than a_c will fail only if the resulting cohesion drop make it unstable. Thus the number and size of landslide on a hillslopes will also depends on local strength, pore pressure and slopes, but we assume that when ground acceleration reaches a_c some minor failures will initiate."

Page 7, paragraphs 2 and 3: This is a good way to define the area affected by landslides that eliminates outliers on slopes having anomalously low critical accelerations. And this should encourage more polygon inventories in the future, which are becoming the norm.

» We hope this criterium can indeed be more robust and push to develop polygon inventories.

Page 9, paragraph 1: Why not use a finite-fault model and examine distances from the

point or area of maximum moment release?

» Page 9 Lines 25-30: A point source is clearly unsatisfying for most cases, and if we want to define the portion of the fault that emits wave it is difficult to define a threshold of moment release. In any case as we state on Page 10 Line 11: "Moreover, cases such as the Niigata or Iwate earthquakes, are still overpredicted when modeled with a single point-source. This suggests that for these cases, with well-constrained source depth, a better prediction of RHMAX is needed, and therefore of either the source term bsat, or the critical acceleration ac."

Page 10, paragraph 3: The statement that better characterization of strength and pore pressure is necessary to refine estimates of critical acceleration is an understatement. Dreyfus et al. (2013) discussed this and should be cited here.

» We write on page 10 Line 23: To define and obtain quantitative estimates of substrate strength or of the ground pore pressure at the landscape scale is an outstanding challenge and lacking relevant constraints, we cannot assess further their influence on the variability of $a\_c$ and $A\_d$. Here, the reviewer thinks to the classical definition of $a\_c$, relating to FS and therefore to pore pressure and strength. But here we question whether the sensitivity of a material to damage (the $a\_c$ we use in this study) varies with pore pressure and material type. It probably does but we can hardly explore it in this paper.

Page 13, paragraph 1: The range of 0.1-0.2 g is not accepted as a "universal acceleration threshold." The Jibson and Harp (2016) study of several of the best documented earthquakes (in terms of landslides) suggests a threshold closer to 0.05 g. The difference is between the outermost limit of the smallest landslides and the zone of concentrated landsliding. This differentiation should be made clearer here. The threshold acceleration values in the different studies are really looking at different landslide limits.

» Page 13 Line 14: We agree. We also note that if 0.15g is a good measure for the concentrated zone of landsliding and 0.05 g is only 3 times smaller for the outer

[Figure]

limits of landsliding. Rather consistent with our assumption that a_c vary moderately compared to Cohesion itself that varies on several order of magnitude. In any case we will write: "It suggests that 0.15g is an appropriate approximation for predicting the area of concentrated landsliding, while the outer limits of landsliding may be rather controlled by a smaller critical acceleration about 0.05 (Jibson and Harp, 2016)."

Figure 1: Define Ro in caption.

» We added in the caption: Ro is the mean depth of wave emission.

Figure 5: Not clear what the red circles indicate.

» Red circles indicates the absolute difference between the modelled maximum distance to wave emission and R95 : If it is positive we overestimate the distance over which are concentrated landslides (by ∼7km for Niigata for example). If it is negative it means significant landsliding persisted further than predicted (by about 5 km for Finisterre, Limon or Wenchuan).

Figure 6: Typo in caption: "name."

» Ok

---

## Referee Comment (RC2) · Anonymous Referee #1 · 13 Mar 2017

The authors did a good job of responding to most of the review comments, but one issue remains a challenge. The authors ascribe a very different definition of "critical acceleration" than has been used historically in the literature. Obviously, they are free to introduce and use any physically reasonable parameters they deem suitable for their analysis. But the term "critical acceleration" and its accompanying notation ($a\_c$) has been in broad usage in the seismic-landslide literature for more than 30 years. This term has a rigorous technical definition that is quite different than the one used by the authors. Adopting a term with a long historical usage and definition and then using it in a completely different context and with a different technical definition is bound to cause a great deal of confusion to knowledgeable readers familiar with the historical usage of the term. And these readers might well draw incorrect conclusions from the paper if they are not familiar with the different definition being used by the authors.

[Figure]

I recommend using a different term for what they are describing in order to minimize the potential for misunderstanding. If that is not possible, at a minimum the authors must clearly define "critical acceleration" on first usage and describe how it differs from the historical usage of the term.

---

## Referee Comment (RC3) · Anonymous Referee #2 · 2 Apr 2017

See attached files

Please also note the supplement to this comment:
http://www.nat-hazards-earth-syst-sci-discuss.net/nhess-2017-71/nhess-2017-71-RC3-supplement.zip

---

## Author Response (AR1)

We thank the two referees for the interest they have shown in the manuscript.

Below we answer in details all their comments and report the new text we have added to clarify some aspect of the manuscript.

In light of these changes and in re-reading the manuscript we have edited some other sentences in the abstract of the main text. All these changes are tracked in a marked version of the manuscript.

Odin Marc, for the rest of the authors,
Patrick Meunier and Niels Hovius.

REPLY TO REFEREE 1

We are pleased by the encouraging and clear review done by referee 1. In this preliminary reply we mainly aim at clarifying the meaning of a_c and why we define it like this.

The Line numbering is following the NHESSD Latex Template, with a Page / Line schemes. We will use it to locate changes related to specific comments. Below the comments are repeated and addressed.

Page 2, paragraph 1: Examining the zone of concentrated landsliding rather than the extreme limits of landsliding is a sound approach to eliminate outliers and unusual conditions.

>> Agreed. We will added on Line 10 Page 2: Moreover, the area with concentrated landsliding should be more closely related to seismic forcing and less dependent on unusual hillslopes conditions.

Page 2, paragraph 2: Suggest adding Keefer (2002) to this list of references. His
updated paper contains additional data.
>> Page 2 Line 12: We now include Keefer 2002.

Page 2, paragraph 2: This relation is parallel to the relation between Arias intensity and
seismic moment developed by Wilson and Keefer (1985, p. 334). This is an early and
somewhat archaic reference, but it laid the groundwork for the kind of modeling done
in the current paper and probably should referenced.
>> Page 2 Line 31:  We now cite (Wilson and Keefer 1985, and Khazai and Sitar 2004) for the link between PGA and landslides, and Meunier 2007 and Yuan 2013 fo rdetailed observations

Page 3, paragraph 2: Unclear what the term "oversteepened slopes" means here.
"Oversteepened" generally means that some geomorphic process has created a slope
having marginal static stability; active cutbanks of rivers are an example this. But earthquakes
trigger landslides on slopes that are perfectly stable in static conditions but that
fail under seismic loading. It is not a matter of oversteepening. And the next line states
that critical acceleration (ac) is independent of slope angle, so why would only oversteepened
slopes be more susceptible to failure? These statements are inconsistent.

>> Page 3 Line 8: By oversteepened we mean here slopes that are steeper than the coefficient of friction of the material and therefore stable only because of a relatively high cohesion. We do not want

to refer to a specific mechanism, and oversteepened is probably misleading.

Our hypothesis is that these "Cohesion stabilized slopes" will fail if the repeated cycles of strong motion cause a sufficient drop of cohesion. Therefore $a_c$ is not the critical acceleration defined using a static force balance and factor of safety, but we define it as a material property, setting the acceleration at which damage (cohesion reduction) initiate. This was more developed in Marc et al 2016b, and we now have developed it here too, to avoid confusion.

 We believe this will resolve most of the recurring problem the referee have underlined with $a_c$ in the next comments. Therefore we have skipped the following comments that only referred to the definition of $a_c$. We keep the notation $a_c$ (to be consistent with Marc et al 2016b) but we define it clearly, state that is is not the classical definition, and later in the text avoid to use "critical acceleration" and use instead "threshold of acceleration for damage".

We will replace: "Over-steepened" slopes by "slopes steeper than the angle of internal friction and thus stable due to a significant cohesion".
First we rewrite: "**they assumed that $a_c$ is a threshold acceleration for damage above which an effective reduction of strength occur...**
Then we will add:
 "Here, we adopt this definition, instead of the classical critical acceleration based on the safety factor and the topographic gradient. With this definition we consider that $a_c$ is much more uniform across a landscape than cohesion which can vary greatly between soil and fractured and intact bedrock. This is consistent with modest variations of the minimum acceleration at which landsliding (Meunier 2007, Hovius and Meunier 2012, Yuan 2013) and minor rockfall (Jibson and Harp 2016) have been observed. The average value of $a_c$ across a landscape is an important constraint on Ad. At the outset, we assume $a_c = 0.15$, which is conservative and allows a focus on areas with significant landsliding. This value is consistent with the observation that non-linear soil behaviour sets in around ground accelerations of around 0.15g (Wen, 1994).
Note, that in this approach, $a_c$ is independent of the slope gradient, but also that a slope experiencing an acceleration larger than $a_c$ will fail only if the resulting drop in cohesion renders it unstable. Thus the number and size of landslides on a hillslope will also depend on local strength, pore pressure and topographic steepness, but we assume that when ground acceleration reaches $a_c$ some minor failures will initiate."

Page 9, paragraph 1: Why not use a finite-fault model and examine distances from the point or area of maximum moment release?

>> Page 9 Lines 25-30: A point source is clearly unsatisfying for most cases, and if we want to define the portion of the fault that emits wave it is difficult to define a threshold of moment release. In any case as we state on Page 10 Line 11:
"Moreover, cases such as the Niigata or Iwate earthquakes, are still overpredicted when modeled with a single point-source. This suggests that for these cases, with well-constrained source depth, a better prediction of RHMAX is needed, and therefore of either the source term bsat, or the critical acceleration ac."

No Change made

Page 10, paragraph 3: The statement that better characterization of strength and pore pressure is necessary to refine estimates of critical acceleration is an understatement. Dreyfus et al. (2013) discussed this and should be cited here.

>> We write on page 10 Line 23: To define and obtain quantitative estimates of substrate strength or of the ground pore pressure at the landscape scale is an outstanding challenge and lacking relevant constraints, we cannot assess further their influence on the variability of a_c and A_d.

Here, the reviewer thinks to the classical definition of a_c, relating to FS and therefore to pore pressure and strength. But here we question whether the sensitivity of a material to damage (the a_c we use in this study) varies with pore pressure and material type. It probably does but we can hardly explore it in this paper., and Dreyfus 2013 is not a proper reference for that.
No changes made.

Page 13, paragraph 1: The range of 0.1-0.2 g is not accepted as a "universal acceleration threshold." The Jibson and Harp (2016) study of several of the best documented earthquakes (in terms of landslides) suggests a threshold closer to 0.05 g. The difference is between the outermost limit of the smallest landslides and the zone of concentrated landsliding. This differentiation should be made clearer here. The threshold acceleration values in the different studies are really looking at different landslide limits.

>> Page 13 Line 14: We agree. We also note that if 0.15g is a good measure for the concentrated zone of landsliding and 0.05 g is only 3 times smaller for the outer limits of landsliding. Rather consistent with our assumption that a_c vary moderately compared to Cohesion itself that may vary across orders of magnitude.

In any case we will write: "It suggests that 0.15g is an appropriate threshold for ground acceleration to predict the area of concentrated landsliding, while the outer limits of landsliding may be rather controlled by a smaller critical acceleration about 0.05 (Jibson and Harp, 2016). However, at lower ground acceleration thresholds, site effect, non-linear attenuation and other secondary controls become increasingly important, disproportionately complicating any prediction of landsliding.."

Figure 1: Define Ro in caption.
>> We added in the caption: Ro is the mean depth of wave emission.

Figure 5: Not clear what the red circles indicate.
>> Red circles indicates the absolute difference between the modelled maximum distance to wave emission and R95 : If it is positive we overestimate the distance over which are concentrated landslides (by ~7km for Niigata for example). If it is negative it means significant landsliding persisted further than predicted (by about 5 km for Finisterre, Limon or Wenchuan).

We added: The absolute error (red circles) indicates the difference in kilometers between the the prediction and the boundary of 95% of the landsliding.

Figure 6: Typo in caption: "name."
>> Corrected.

REPLY TO REFEREE 2
We thank referee 2 for its in-depth examination of the model development.
Below we address his comments and the modifications we adopted to be more consistent.

This manuscript reports an interesting attempt to define a model to predict area affected by earthquake-induced landslides, outlining distance from earthquake source, within which major effects are expected, on the basis of seismological parameters. While the basic ideas developed to simplify the calculation of such distance appear smart, some aspects of model implementation seem to me unclear or questionable and should be better justified or reconsidered.

A first problem concerns the equation (3) used to define the relation between the seismic moment $Mo$ and the fault rupture length $L$, i.e.:

$$L = \frac{Mo^{\frac{2}{5}}}{\mu C_1^{\frac{3}{2}} C_2} \quad \text{[R1]},$$

where $\mu$ is the rigidity modulus of the faulted rocks and $C_1, C_2$ are empirically determined coefficients. The authors declare to have derived it from the paper by Leonard (2010). However the cited paper does not report a relation $L(Mo)$ in this form, and, if equation (3) was derived from the results presented by Leonard, it is incorrectly written.

Indeed Leonard, starting from the well known general equation
$$Mo = \mu \, LW \acute{D} \quad \text{[R2]},$$
where $W$ is the fault rupture width and $\acute{D}$ is the mean dislocation along the rupture fault, proposes two equations relating $W$ and $\acute{D}$ to $L$, in the forms
$$W = C_1 L^{\beta} \quad \text{[R3]},$$

$$\acute{D} = C_2 (LW)^{\frac{1}{2}} = C_2 \left[ C_1 L^{(1+\beta)} \right]^{\frac{1}{2}} \quad \text{[R4]},$$

from which one can obtain
$$Mo = \mu C_1^{\frac{3}{2}} C_2 L^{\frac{3}{2}(1+\beta)} \quad \text{[R5]}.$$
Leonard found that, for almost all kinds of fault, $\beta$ can be set to 2/3, which implies
$$Mo = \mu C_1^{\frac{3}{2}} C_2 L^{\frac{5}{2}} \quad \text{[R6]},$$
with the exception of strike-slip faults exceeding a length of 45 km, for which $\beta$ should be set to 0 and consequently.

$$Mo = \mu C_1^{\frac{3}{2}} C_2 L^{\frac{3}{2}} \quad \text{[R7]}.$$
From these equations, one can derive that, for most of faults,
$$L = \left( \frac{Mo}{\mu C_1^{\frac{3}{2}} C_2} \right)^{\frac{2}{5}} \quad \text{[R8]},$$
(which differs from [R11]) and, for strike-slip faults longer than 45 km,
$$L = \left( \frac{Mo}{\mu C_1^{\frac{3}{2}} C_2} \right)^{\frac{2}{3}} \quad \text{[R9]}.$$

Additionally, Leonard derived different values of $C_1$ and $C_2$, for different type of faults, i.e., $C_1 = 17.5$ and $C_2, = 3.8 \cdot 10^{-5}$ for dip-slip inter-plate faults, $C_1 = 15.0$ and $C_2, = 3.7 \cdot 10^{-5}$ for strike-slip inter-plate faults and $C_1 = 13.5$ and $C_2, = 7.3 \cdot 10^{-5}$ for intra-plate earthquake (stable continental regions). The value

assumed for $C_l$ in the present manuscript (16.5) does not correspond to none of the values proposed by Leonard and also the value assumed for $\mu$ (3.3 GPa) is incorrect (it should be 33 GPa). If the errors in equation form and in parameters were due to misprints, they should be corrected, but if these formulae were actually used in calculations, the results would be totally inconsistent with the seismological model and should be recalculated.

>>      The derivation reminded here is exact and we have followed the same. The script we used to compute rupture length indeed use EQ 8 and EQ 9 (where C2 = 17km in EQ 2). Indeed, (Mo^2/5) / $\mu$ C2 C1^3/2 would have given ridiculously small length on the order of some meters...

→ We have added the missing parentheses before the exponent in the equation 3.

For the parameters, we used indeed μ=33 Gpa,  and the dot is a typo → corrected.

For C1, 16.5 is an intermediate value between 15 and 17.5 (and well within the 1sigma of both estimation that spans approximately between 10 and 25) that allow to collapse strike-slip and reverse fault before they reach the seismogenic depth. We note that using 15 or 17.5 or strike slipe dip slip would change by +6% / - 3% (respectively) the rupture length prediction having a quantitatively negligible effect. Similarly we use a single C2 value 3.7.10^-5 and not 3.7 and 3.8.
We somehow overlooked that and did not state it in the text. We are sorry about that and now have expanded the sentence after Eq 3:

"with mu the shear modulus, assumed to be 33 GPa, and C_1 and C_2 empirical constants. Although Leonard (2010) fitted independently strike slip and dip-slip faults he obtained similar values in both cases, C_1 = 15 [11-20] and C1=17.5 [12-25] , and C2 = 3.8 and 3.7 10-5, respectively. For the sake of simplicity and to have a single prediction for strike-slip and reverse faults with small and intermediate lengths, we choose intermediate values and use C1=16.5 m^1/3 and C2=3.7.10^-5. With this assumption, predicted lengths differ only by a few per cent from what would be obtained with the best estimate proposed by Leonard and reported above."

Another puzzling question is relative to the equations (4), i.e.

$$b = b_{sat} \exp\left[ e_5\left(M_W - M_h\right) + e_6\left(M_W - M_h\right)^2 \right] \qquad \text{(for } M_W \le M_h) \qquad [R10]$$
$$b = b_{sat} \exp\left[ e_7\left(M_W - M_h\right) \right] \qquad\qquad \text{(for } M_W > M_h),$$

which were used to define the peak ground acceleration (PGA) expected at a distance of 1 km for an event of magnitude $M_W$. This acceleration value, in turn, is used to derive the distance $R_{HMAX}$ within which the ground acceleration is not less than $a_c$ (assuming that ground motion attenuation depends only on geometrical spreading), according to the equation

$$R_{HMAX} = \sqrt{\left(\frac{b}{a_c}\right)^2 - R_o^2} \qquad [R11].$$

The authors declares to have based their calculations on the ground motion prediction equation (GMPE) proposed by Boore & Atkinson (2008), but they adopt an arbitrary value of 4000 m for $b_{sat}$, which properly should be defined as the acceleration at a distance of 1 km for an event of magnitude

$M_W$ equal to the magnitude "hinge value" $M_h = 6.75$.

Preliminarily, I observe that it is quite puzzling to propose, for an acceleration, a value measured in meters. Probably the misunderstanding about the meaning of $b_{sat}$ derives by the fact that $b$ is used to calculate the distance where acceleration is reduced to $a_c$, exploiting the inverse proportionality between wave amplitude and distance. Actually, following the GMPE model by Boore & Atkinson, $b_{sat}$ should be defined as the acceleration expected for $M_W = M_h$ at a reference distance $R_{ref}$, which Boore & Atkinson set to 1 km. Indeed, the complete expression of Boore & Atkinson's GMPE would include a factor depending on distance $R$ which becomes equak to 1 when $R = R_{ref}$. Thus, to avoid a dimensional inconsistence, [R11] should be written as

$$R_{HMAX} = \sqrt{\left(\frac{b}{a_c} R_{ref}\right)^2 - R_o^2} \qquad \text{[R12]}.$$

Numerically [R12] gives the same result as [R11] only if distances are expressed in km, but in any case the equation [R12] is dimensionally correct, assuming that both $b$ and $a_c$ represent accelerations. It is however unclear while, adopting the Boore & Atkinson's GMPE, the authors did not simply derives $b_{sat}$ from it. Indeed, this GMPE provides the element to calculate $b_{sat}$ for different type of faults, in terms of expressions like $\exp(e_1)$ for unknown type, $\exp(e_2)$ for strike-slip, $\exp(e_3)$ for normal faults and $\exp(e_4)$ for reverse faults, where the coefficients $e_1$, $e_2$, $e_3$ and $e_4$ are reported in Table 7 of the cited paper. Furthermore, the author, using equation (4) ([R11] in the present comments), report to have set coefficients $e_5 = 0.6728$, $e_6 = -0.1826$ and $e_7 = 0.054$, assuming that these provide ground acceleration at 1 Hz. Actually, these coefficient values appear derived from those reported by Boore & Atkinson for 5% damped pseudo-spectral accelerations at a period of 1 s (apart from a slight error in $e_5$ which actually is 0.6788: see Table 7 in Boore & Atkinson, 2008). These coefficients are relative to GMPE that does not predict ground motion, but the response of a one degree-of-freedom oscillator whose base is fixed to soil and forced to move by seismic ground motion. This shaking parameter is used to evaluate the response of engineering structures (which can be assimilate to an oscillator of given eigen-frequency and damping) in terms of maximum acceleration induced by seismic shaking to the oscillator. It seems to me hardly justifiable to assimilate slope material behaviour to an oscillator with eigen-frequency of 1 Hz and damping equal to 5% of the critical values (which is typical for quite elastic engineering structures). Thus, I wonder why it was not simply used the coefficients provided for PGA in the same Table 7 (which, actually, predict a saturation for $M_W \geq M_h$ as resulting from being $e_7 = 0$)?

>> What is above is correct but not contradictory with our approach, although it may have been awkwardly presented.

**1/ On the unit/definition of b:**
 b and a_c should be non-dimensional acceleration, normalized by the gravitational acceleration, g. Thus we should get a normalized ground acceleration at the surface when dividing b by a depth, for example, b/R0 = 0.4 , with b=4000 m and R0=10km. That is 40% of g.

Anyway for clarity, **we rewrote Line 27 Page 3** with the equation suggested and with: " with b the non-dimensional near-source acceleration at a reference distance from the seismic source, Rref = 1000 m, and R0 ... " and included this modifications in Eq 4 and 6.
**We also added at Line 1 Page 3**: "Note that here, a_c and all other accelerations are normalized by the gravitational acceleration and thus non-dimensional."

**2/ On the magnitude scaling and the choice of b_sat**
The main misunderstanding is that we do not aim at reproducing Boore and Atkinson 2008 prediction, which depends on a number of term that we do not consider and/or cannot constrain, such as Vs30 for site effects and non linear attenuation terms.

This should be better stated on the text, but we simply aim at a ground shaking model where reasonable scaling for magnitude, fault type and attenuation are used.
For fault type: based on Boore and Atkinson shaking is ~ 30% lower for Normal fault than for strike-slip/dip-slip; As stated in the text.
For Attenuation: for frequency around 1Hz, geometrical spreading will dominate on the ~30 first km, that are the typical distance from the rupture where landslide occur. As stated in the text.
Then for Magnitude we aim at reproducing a increase of shaking and then saturation of the shaking consistent with seismological observations for earthquakes.

**Line 30 Page 3 We rewrite:** "To construct a simplified ground shaking model that is consistent with first order seismological observations and landslide triggering we use the scaling of the near-source acceleration, $b$, with earthquake magnitude proposed by Boore and Atkinson (2008)":

**Line 14 Page 4 we added:** "We refrain from using b_sat values as derived by Boore and Atkinson (2008) because they should be included within a model accounting for site effects and non-linear attenuation, attributes that are beyond the scope of this work. Therefore, to be consistent with the model of Marc et al. 2016 we use ..."

Additionally we also added to make it clear to the reader, that although we assume a single value for b_sat we discuss its potential (and likely) variations later in the manuscript:
**We rewrite Line 17 Page 4:**
 "If a_c **and b_{sat}** are relatively constant as suggested by Marc et al., 2016b then the model is fully constrained without any free parameter. The assumptions that $a\_c$ **or $b\_{sat}$ are constant** across all settings are discussed in Section 5."

**3/ On the scaling parametrization**
First of, the wrong value of e5 is a typo, and we used the correct one for our calculations. Then we rewrote: "e5,e6 and e7 are empirical constants for 1Hz Pseudo Spectral Accelerations."

Then we agree that PSA are not likely a correct mechanical description for hillslopes failure but they have the advantage to be available at different frequency. Because different frequency-dependent processes modulates the ground shaking (source spectra, non-linear attenuation, cf Boore 2013), we do see different behavior before and after saturation (when M<Mh or M>Mh) for PSA at different frequency (Boore and Atkinson 2008). Thus we prefer to use a scaling consistent with the 1Hz frequency, while PGA scaling is closer of PSA ~5-10Hz, with much less decay between PGA at saturation and PGA for a Mw = 5. As we currently state in the text, typical frequency of resonance of hillslope are around 1 Hz (Meunier et al., 2008) while smaller structures prone to failure maybe most sensitive to higher frequencies, up to 5-10 Hz (Line 34 page 3, Marc et al 2016b)

Therefore we added Line 15 Page 4 : "We prefer to use the scaling of pseudo-spectral accelerations (rather than the one for peak ground accelerations for example) because it focuses on ground motion

around a specific frequency, and therefore should better represent the frequency-dependent modulation of the ground shaking. The mechanical assumptions associated with pseudo-spectral accelerations (elastic oscillator with base fixed on the ground and 5% damping) are probably not entirely adequate for hillslope failure, but not inconsistent with the resonance of  topographic ridges themselves (Meunier et al., 2008)."

Other minor comments relative to specific points of the manuscript can be found highlighted in the enclosed pdf copy.

We have rewritten all words/sentences/references highlighted by the referee (mainly minor writing issues and typos).

**Page 2  Line 20** : Subscript at Log10;

**Page 2 Eq 1**: replace equivalent by proportional in Eq 1

**Page 2 line 11:** The a and b  for multi paper in the same years are defined based on the reference list sorting (according to NHESS / Copernicus guidelines). Thus, it seems that Marc et al., 2016b, may appear first in the text.

**Page 3 Line 33** This Meunier 2013 is now Meunier et al., 2013b, the other one (Page 12) is Meunier et al., 2013a

**Page  4 Line 29 / 30**: We wrote :  "Note that Ctopo cannot be computed as the fraction of As within which slopes are less than 10° because local flats will impede landsliding and change the landslide density, but rarely the distribution area defined as an envelope containing all earthquake-induced landslides, independent of variations of landslide density."

To clarify we rewrite: Note that Ctopo cannot be computed as the fraction of As within which slopes are less than 10° . This is because local flats will impede landsliding and change the landslide density locally, but not necessarily reduce the envelope containing all earthquake-induced landslides, that typically defines Ad."

The idea is that small area of flatlands at shorter distance from the fault than  RHMAX will not affect the fringes of the decaying landslide density that are typically used to define and measure A_d.

**Page 7  Line 17 :** Aa changed to A_d

**Page 2  Line 25 :** We rewrite: "with the with the apriori predicted length, **based on the scaling proposed by Leonard (2010), ...**"

**Page 2  Line 33  :**  We now refer to Figure 2 and 3

**Page 9  Line 15:** We originally wrote : "but that it sometimes overpredicts the affected area (when 100% of the landslides are within the model distribution area"

It is true that we do not compare yet to the R95 and that for Aysen and Niigata the model do not capture 100% of the landslides. But still we can argue that Iwate and Niigata are overpredicted, with the model limits well beyond the last mapped landslide in most directions, (Figure 3). We do not pronounce ourself here on overprediction for Aysen that is less clear.

Finally to be clearer we rewrite in the text : "but that it sometimes overpredicts the affected area, with its limit well beyond the outermost mapped landslide in most directions, as for the Niigata, and Iwate cases (Figure 3)"

**Page 10  EQ 6 :** Brackets added and several formatting errors were corrected.

**Page 12  Line 30:** We corrected the sentence to: " For Epi_as < 0.4 we observed ..."

**Figure 2 Caption :** typo corrected: strike

**Figure 6 Caption :** typo corrected: name

[revised manuscript text omitted]

---

## Author Response (AR2)

We are pleased that our manuscript can be published in NHESS. I have corrected the few typos in the main manuscript. The marked PDF versions follow.

Odin MARC

In my opinion, after this revision, the manuscript can be accepted for publication on NHESS.
Below I list just a few supplementary indications for minor editing (mainly relative to typos).

Page 1 - line 2: "landslidestriggered".

line 8: "seimsic".

Page 3 - line 25; "onlyif".

Page 4 - line 16: Put a comma after "dip-slip faults".

Page 5 - lines 18-20: Replying to my comment relative to Page 4 - lines 29-30 of the first submission, the authors declare to have modified this sentence to make it clearer, rewriting
"Note that Ctopo cannot be computed as the fraction of As within which slopes are less than 10°. This is because local flats will impede landsliding and change the landslide density locally, but not necessarily reduce the envelope containing all earthquake-induced landslides, that typically defines Ad."
However, this change was not reported in the revised version of the manuscript.
>> This was an error. The new sentence is now in.

Page 6 - line 23: Close bracket is missing after "(Table 1".
- line 26: "a limited areas"

Page 8 - line 11-12: "(Figure 2,3".

Page 12 - line 33: "groun".

[revised manuscript text omitted]